# LEMONS: An open-source platform to generate non-circular, anthropometry-based pedestrian shapes and simulate their mechanical interactions in two dimensions

Oscar Dufour[1][*], Maxime Stapelle[1][‡], Alexandre Nicolas[1][†]

**1** Université Claude Bernard Lyon 1, Institut Lumière Matière, CNRS, UMR 5306, 69100, Villeurbanne, France

* oscar.dufour@univ-lyon1.fr ,    ‡ maxime.stapelle@univ-lyon1.fr ,        † alexandre.nicolas@cnrs.fr

## Abstract

To model dense crowds, the usual recourse to oversimplified (circular) pedestrian shapes and contact forces shows limitations. To help modellers overcome these limitations, we propose an open-source numerical tool. **It consists of a Python library that generates 2D and 3D pedestrian crowds based on anthropometric data, and a C++ library that computes mechanical contacts with other agents and with obstacles, and evolves the crowd's configuration. Additionally, we provide an online platform with a user-friendly graphical interface for the Python library, and scripts to call the C++ library from Python. The tool enables users to implement their own decisional layer, i.e., to control the agents' choices of desired velocities.**

# 1   Introduction

## 1.1   Motivations

From an external physical viewpoint, a pedestrian is a deformable mechanical body. As such, the pedestrians' bodies obey Newtonian mechanics. In particular, they experience physical forces (e.g., if they happen to push against a wall) that partly constrain their motion. Yet, they differ from inert objects in that, upon flexing their muscles, they can *internally* deform their bodies and thus self-propel via the interaction with the ground. This reveals two intrinsically coupled levels of pedestrian dynamics: the mechanical level and the decision-making level (which controls the internal body deformation and is not governed by Mechanics). The literature on crowds reflects this duality. Some studies focus on mechanical aspects (essential in high-density scenarios) [1, 2] but most often relying on idealised interaction forces and simplified circular shapes that fail to replicate mechanical interactions faithfully; others examine decision-making (especially relevant in low-density contexts) [3, 4], while yet others [5] address the coupling of both levels, crucial in intermediate-density situations where individuals navigate to avoid collisions, but may nonetheless experience physical contact.

Most existing crowd dynamics models [2, 5, 7] represent pedestrians as disks. However, when using the bideltoid breadth (defined in Fig. 2) as the disk diameter, a tightly packed random arrangement of a realistic population (shown in Fig. 1) only achieves densities of about $4\,\mathrm{ped/m^2}$. This falls far short of empirically observed peak densities, which sometimes exceed $8\,\mathrm{ped/m^2}$ in real-world scenarios [8–10]. One idea to reconcile this density discrepancy could be to reduce disk diameters. However, this adjustment introduces critical flaws. First, it preserves the unrealistic circular body geometry, which fails to reflect human morphology and limits the number of simultaneous physical contacts a pedestrian can have to at most six. In contrast, in controlled dense crowd scenarios [11, 12], single individuals

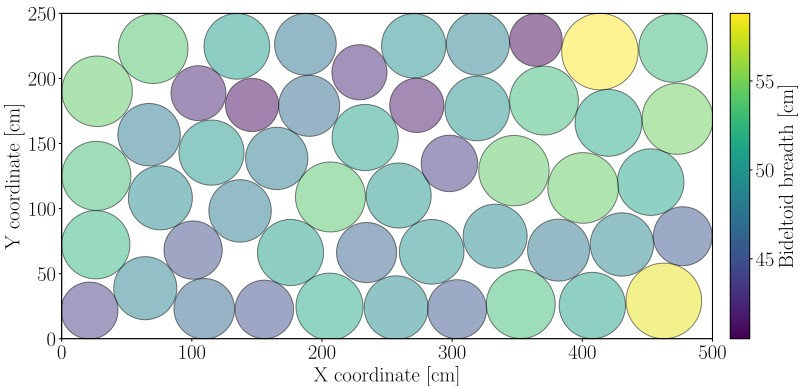

Figure 1: Tightly packed random pedestrian disk arrangement, reaching a density of $4\,\mathrm{ped/m^2}$. The disk diameters are sampled from the empirical bideltoid breadth distribution of a US population subset (ANSURII database, [6]), with mean 49 cm and standard deviation 4 cm. Algorithm details: App. D.

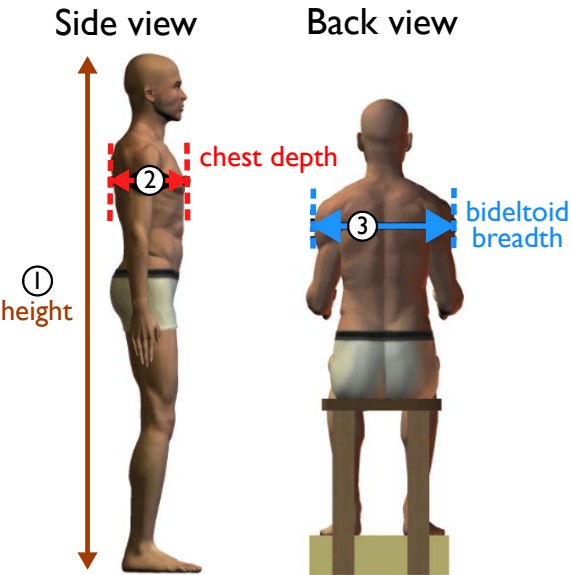

Figure 2: Illustration of anthropometric measurements – including height, chest depth and bideltoid breadth – adapted from [6].

57 experiencing simultaneous contact with eight distinct others were observed. Second,
58 narrower disks would artificially lift constraints on unidirectional flow in narrow corridors
59 and overestimate the associated flow rate. Therefore, instead of reducing disk diameters, we
60 refine existing elongated-body formulations and represent the mechanical shape of a
61 pedestrian by an anisotropic shape that better captures pedestrian morphology and
62 multi-contact interactions.
63 The use of anisotropic shapes in the granular materials literature is well-established.
64 Discrete element simulations have employed diverse geometries to describe solid dynamics:
65 ellipses [13], polygons [14], polar-form polygons [15], and disk assemblies [16] represent
66 key examples, while [17] provides a comprehensive review. Despite extensive research on
67 granular dynamics, non-circular body shapes have so far only been integrated into a
68 relatively small number of pedestrian models. Elliptical volume exclusion was incorporated
69 into generalised centrifugal force models, prioritising inertial forces over traditional
70 damped-spring mechanics for pedestrian contact [18]. For models relying on the concept of

velocity obstacles, the original circular agents' shapes were gradually extended to ellipsoids (EORCA), or polygonal approximations thereof for computational efficiency [19, 20], and to arbitrary shapes approximated by stitching rounded trapezoids centred on the medial axis of the shape [21]. These ellipsoid arbitrarily shaped representations govern the choice of an optimal velocity that ideally enables collision-free navigation (decisional purpose), alien to any consideration of mechanical interactions. If one focuses on models including short-range and/or contact interactions, Langston et al. represented pedestrians with three overlapping circles in a discrete-element simulation [22]. So, too, did Korhonen et al. (FDS+EVAC) [23] and Song et al. [24] in the mechanically simpler context of modified social-force models (also see the 1995 paper by Thompson et al. [25]), while spheropolygons [26] or spherocylinders [27, 28] were later introduced in force-based simulations incorporating self-propulsion forces as well as granular material interactions governed by Newtonian mechanics, notably to model competitive egress scenarios. Recently, the human torso has been modelled as a capsule in a flow governed by position-based dynamics, supplemented with short-range interactions [29].

Nevertheless, albeit anisotropic, these shapes face significant limitations: they are more or less arbitrarily defined and lack a quantitative medical or anthropometric basis. Consequently, the generated crowds lack representative heterogeneity, which is crucial for accurately replicating density and collision statistics. These rigid structures also resist extension to new contact models involving deformation or relative motion between the centres of mass of the body segments. This article addresses these limitations by introducing a tool that generates realistic crowds from anthropometric data, simulates mechanical interactions, and allows user-defined decisional layers. It therefore removes the technical barriers when it comes to modelling elongated crowd shapes, allowing the community to focus on decision-making and its interaction with mechanics. This tool also opens the door to exploring essential questions about introducing complexity into modelling, such as whether one needs to introduce a third dimension or incorporate heterogeneity in agent types, like strollers or individuals carrying bags.

In addition to serving researchers in the field, this tool is designed for crowd modellers at all levels, starting from beginners, in their efforts to assist, e.g. public authorities and businesses; it provides them with the possibility to achieve more realistic simulations of dense (and possibly heterogeneous) crowds in a simple way. In this particular regard, existing simulation software[1], such as Iventis [30] and Vadere [31], fail to reflect the latest advances; our solution brings crowd simulation into the present.

Finally, the proposed tool, dubbed LEMONS, may also be of pedagogical interest. It can easily be integrated into classroom settings, enabling teachers and science communicators to simulate agents with minimal effort. It has the potential to spark interest in the physical sciences, particularly in the study of complex systems and active matter.

## 1.2  How to read this document

This document exposes the theoretical foundations of the LEMONS software tool and provides an overview of the code structure. A minimal usage example, detailed usage tutorials (along with Jupyter Notebooks), and comprehensive API documentation for the classes and functions in LEMONS are available online [32]. Great care has been taken in developing the codebase to make it user-friendly, easy to expand, and maintainable, as detailed in App. E.

This article is structured as follows. We begin by outlining the theoretical foundations of the project, introducing a novel mechanical shape for pedestrians, and describing the

---

[1]The GitHub repository https://github.com/pozapas/awesome-crowdynamics aspires to compile a broad collection of existing crowd and pedestrian open source simulation software.

generation of realistic crowds based on anthropometric data. Sec. 2.2 also details the specification of mechanical interactions between agents' shapes and with any walls present in the environment. The document then provides an overview of the code structure in Sec. 3. Finally, in Sec. 4, we present an in-depth discussion of our model, outline the tests conducted to validate its implementation, and propose potential directions for future improvements. We also provide detailed instructions for running a pushing scenario simulation in this section. Supplemental videos of the tests and of the practical case studied are also provided [33].

## 2 Theory & Methods

### 2.1 From the individual pedestrian's shape to the generation of a synthetic crowd

For realistic pedestrian shapes, we relied on medical data, specifically, cross-sectional images from two cryopreserved middle-aged cadavers (a male at 1 mm intervals and a female at 0.33 mm intervals) provided by the Visible Human Project [34]. Since 2D simulations prevail in the field of pedestrian dynamics, we need to project the 3D shape onto a suitable effective 2D shape. To this end, we selected the cross-section at torso height[2], an example of which is shown in Fig. 3 for the male specimen; this choice is notably justified by the fact that fatalities during crowd crushes often result from asphyxia and severe compression of the rib cage and lungs. We approximated the torso slice with a set of five partly overlapping disks: two for the shoulders, two for the pectoral muscles, and one for the back, as illustrated in Fig. 7. Disks were chosen over polygons because defining and computing mechanical contact between disks is much simpler and more computationally efficient [17].

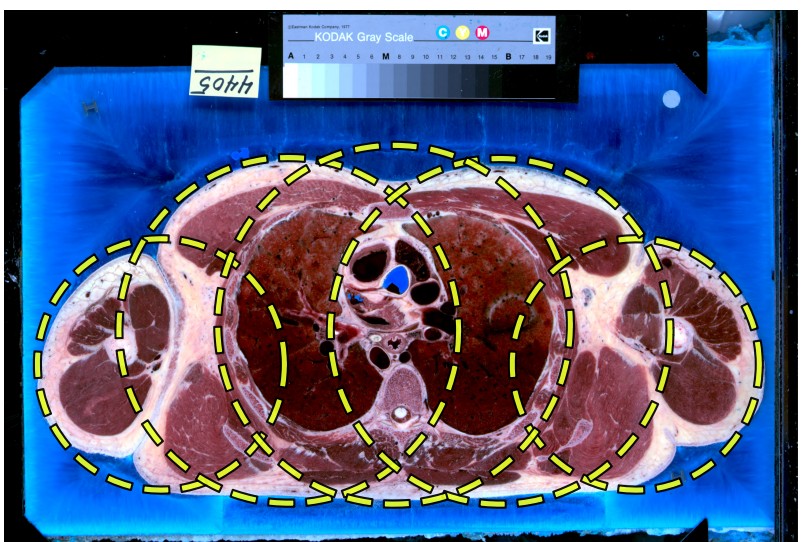

Figure 3: Torso section of a cryopreserved man, slice number 4405, from the [34] database, covered with five disks. The 'Kodak Q-13 Gray Scale' ruler measures 20.3 cm by 2.5 cm.

To extend the fitting method to a whole population, we utilised anthropometric data from the ANSURII database [6], which comprises 93 body measurements from 6,000 US

---

[2]The torso height corresponds to the height where chest depth and bideltoid breadth are measured in the anthropometric data, as shown in Fig. 2 from the Visible Human Project cadavers. Specifically, it corresponds to 151.6 cm for the male cadaver and 138.369 cm for the female cadaver.

Army personnel (4,082 men and 1,918 women). In the Anthropometry tab of our online app, these data are easily accessible, viewable, and downloadable. Note, however, that this sample is not fully representative of the US civilian population; in particular, among other selection biases, men are over-represented, whereas women form the majority of the US population, according to the NHANES database [35] [3] (which can be partly explained by the higher life expectancy of women in the US population). To generate a crowd that reflects the anthropometric diversity of ANSURII starting from the foregoing 2D projection made of five disks, we translate the centres of the disks with a homothety centred at the pedestrian's centre of mass and scale their radii to match empirical chest depth and bideltoid breadth measurements (defined in Fig. 2). These geometric operations do not perfectly preserve the initial shape, but they achieve a realistic approximation. Compared to the maximal possible density around $4\,\mathrm{ped/m^2}$ for circular agents (see Fig. 1), crowds generated with these methods can reach a density of $7.2\,\mathrm{ped/m^2}$ (see Fig. 4), much closer to empirical measurements in very dense situations [8–10].

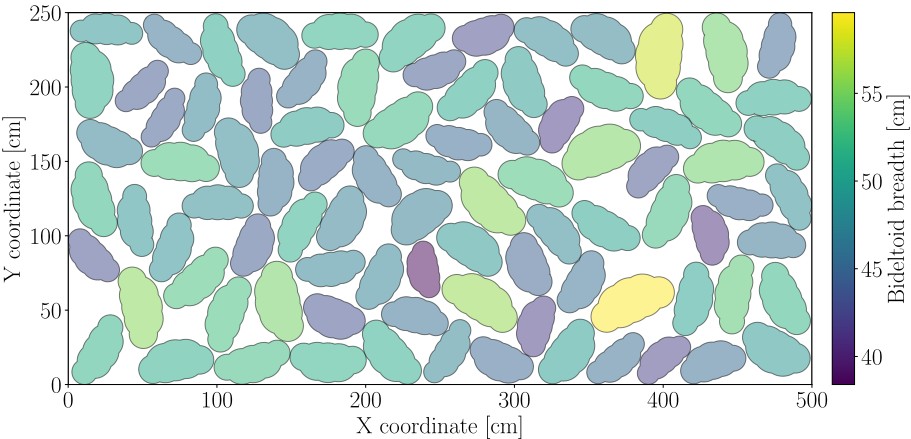

Figure 4: Tight random packing of pedestrians without preferred orientation using an arrangement of five disks, reaching a density of $7.2\,\mathrm{ped/m^2}$. Both the sample from the ANSURII database [6] and our model database have a mean bideltoid breadth of 49 cm and a mean chest depth of 25 cm.

## 2.2 Mechanical interactions

Studying mechanical interactions between pedestrians is inherently complex, owing to factors such as three-dimensional contact geometry, protective hand movements during contacts or falls, and non-static, multi-point contact configurations [12, 36]. Biomechanical studies on both embalmed and unembalmed (fresh, non-rigid) cadavers subjected to dynamic loading – both frontal [37, 38] and lateral [39] – have characterized thoracic impact response, derived the Lobdell mechanical model for the human thorax under blunt impact [40], and subsequently refined and extended it [41, 42]. In addition, contact forces between pedestrians have been measured under varying degrees of crowding, in both static and dynamic conditions [43]. Nonetheless, the fundamental nature of live pedestrian-to-pedestrian contact remains poorly understood, particularly during complex, multi-body collisions in which active responses, such as the use of the hands, can substantially influence the interaction dynamics. To render the problem tractable, we therefore simplify these interactions by relying on granular material interactions between the disks that constitute each agent's shape. Specifically, we model the

---

[3]NHANES provides only limited measurements and lacks key metrics such as bideltoid breadth and chest depth, and therefore cannot be used in our software.

interaction in the simplest way we find appropriate, using a single-damped spring as illustrated in Fig. 5:

- In the normal direction (orthogonal to the contact surface), the interaction is described by a spring in parallel with a dashpot (Kelvin–Voigt model), which captures both elastic effects and energy dissipation;
- In the tangential direction (parallel to the contact surface), the interaction is modelled by a parallel spring-dashpot system in series with a slider, reflecting Coulomb's law. This slider represents a threshold-based element that resists tangential motion until a critical force threshold, proportional to the normal force, is exceeded; after this threshold is reached, it slips at a constant force.

Another force is introduced to encompass the effective backwards friction with the ground over a step cycle, controlled by the deformation of the body. Technical details are given in App. A.1, and a comprehensive overview of notations, definitions, and mathematical expressions can be found in App. A.3.

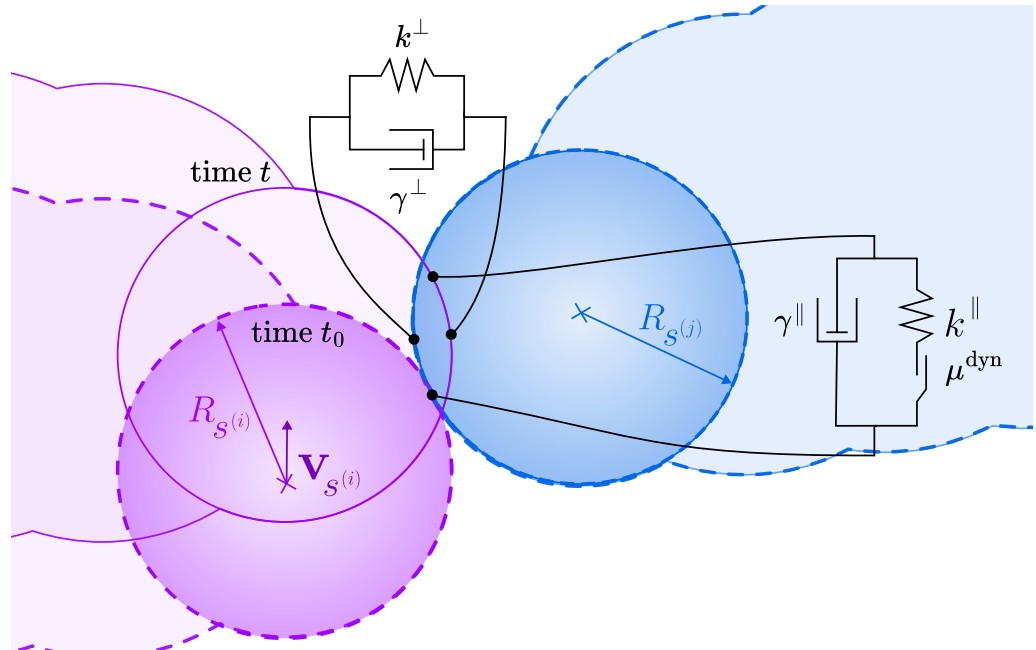

Figure 5: Interactions between composite disks of pedestrians $i$ (radius $R_{s^{(i)}}$, velocity $\mathbf{v}_{s^{(i)}}$) and $j$ (radius $R_{s^{(j)}}$, stationary with $\mathbf{v}_{s^{(j)}} = \mathbf{0}$) are modeled using mechanical elements.

Finally, the deliberate forward motion is subsumed into a propulsion force $\mathbf{F}_p$ for translational motion (and a propulsion torque $\tau_p$ for deliberate rotations of the torso) which result from the pedestrian's decision-making process. The LEMONS platform is agnostic to the decision-making model: it expects the user to define $\mathbf{F}_p$ and $\tau_p$ for each agent as they see fit (see Eq. 4 for a crude proposal). The equation of motion of the centre of mass of agent $i$ (mass $m_i$, translational velocity $\mathbf{v}_i$) is then expressed as:

$$m_i \frac{d\mathbf{v}_i}{dt} = \mathbf{F}_p - m_i \frac{\mathbf{v}_i}{t^{(transl)}} + \sum_{(s^{(i)}, s^{(j)}) \in \mathcal{C}_i^{(ped)}} \left( \mathbf{F}_{s^{(j)} \to s^{(i)}}^{\parallel contact} + \mathbf{F}_{s^{(j)} \to s^{(i)}}^{\perp contact} \right)$$
$$+ \sum_{(s^{(i)}, w) \in \mathcal{C}_i^{(wall)}} \left( \mathbf{F}_{w \to s^{(i)}}^{\parallel contact} + \mathbf{F}_{w \to s^{(i)}}^{\perp contact} \right) \tag{1}$$

where

$$
\begin{aligned}
\mathcal{C}_i^{(\text{ped})} &= \left\{ (s^{(i)}, s^{(j)}) \,|\, s^{(j)} \text{ in contact with } s^{(i)} \right\}, \\
\mathcal{C}_i^{(\text{wall})} &= \left\{ (s^{(i)}, w) \,|\, w \text{ in contact with } s^{(i)} \right\}.
\end{aligned}
\tag{2}
$$

Here, the symbol $\|$ indicates a force tangential to the contact surface, while $\perp$ signifies a force orthogonal to the contact surface. $t^{(\text{transl})}$ is a characteristic timescale for the effective backwards friction and the $s^{(i)}$ represents the five disks that form agent $i$. Importantly, depending on the time scale $t^{(\text{transl})}$, Eq. 1 will describe either inertial, underdamped translation dynamics (long $t^{(\text{transl})}$) or overdamped dynamics (short $t^{(\text{transl})}$).

These forces are applied at the contact centres to induce torques on the torso. The rotational dynamics of agent $i$'s torso (moment of inertia $I_i$, angular velocity $\omega_i$) are governed by:

$$
I_i \frac{d\omega_i}{dt} = \tau_p - I_i \frac{\omega_i}{t^{(\text{rot})}} + \sum_{(s^{(j)}, s^{(i)}) \,\in\, \mathcal{C}_i^{(\text{ped})}} \tau_{G_i,\, s^{(j)} \to s^{(i)}}
+ \sum_{(s^{(j)}, s^{(i)}) \,\in\, \mathcal{C}_i^{(\text{wall})}} \tau_{G_i,\, w \to s^{(i)}}
\tag{3}
$$

where $t^{(\text{rot})}$ is a characteristic timescale for rotational damping and the $\tau_{G_i,\, s^{(j)} \to s^{(i)}}$ refer to the torque at the centre of mass $G_i$ of pedestrian $i$, resulting from pedestrian-pedestrian interaction forces. A (very) crude choice for the propulsion force and torque is

$$
\mathbf{F}_p = m_i \frac{v^{(0)} \mathbf{e}^{(\text{target})}}{t^{(\text{transl})}} \quad \text{and} \quad \tau_p = I_i \frac{-\delta\theta}{(t^{(\text{rot})})^2},
\tag{4}
$$

where $v^{(0)}$, $\mathbf{e}^{(\text{target})}$, and $\delta\theta$ are the preferential speed, the unit vector pointing to the target (or way-point), and the angular mismatch between the target direction $\mathbf{e}^{(\text{target})}$ and the front direction of the body of agent $i$. To solve this set of coupled differential equations of motion, we employed the standard Velocity-Verlet algorithm[4] mentioned in [44], section 3.

## 3 The Codebase

The software release consists of (i) an online platform https://lemons.streamlit.app/ to generate and visualise individual pedestrians (whose shapes are compatible with anthropometric data) or crowds, (ii) a C++ library to compute mechanical contact forces in two dimensions and then evolve the crowd according to Newton's equation of motion (in the overdamped regime or in the inertial, underdamped one), and (iii) a Python interface to import anthropometric data, generate and visualise crowds, and simulate their dynamics via simple calls to the C++ library. It introduces a generic configuration file format (stored as XML) to store agents' shapes and mechanical properties, as well as crowd configurations.

### 3.1 XML crowd configuration classes

Several levels of detail must be specified to define the configuration of a (presently 2D) crowd, from the geometric and mechanical properties of each of its agents to their positions. We introduce a generic structure composed of nested classes, stored as XML files (processed by the third-party library `TinyXML-2`), included in the codebase, to mimic these levels of

---

[4]To save computational time and avoid looping over all agents and walls at each time step of the simulation to determine $\mathcal{C}_i^{(\text{ped})}$ and $\mathcal{C}_i^{(\text{wall})}$, the algorithm's implementation relies on a careful definition and handling of neighbour lists; see App. B for details of our neighbour-determination procedure.

information; we hope this structure will be used broadly for the definition of crowd configurations. To illustrate its generality, alongside standard adult pedestrians, we will also instantiate geometric objects corresponding to cyclists on bikes.

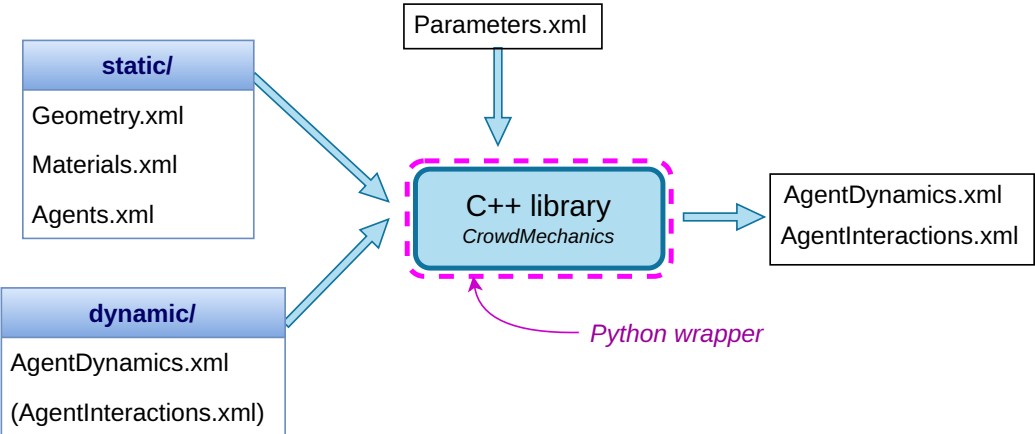

Figure 6: Functional diagram showing the XML configuration files defining the crowd and used as input and output of the mechanical simulation routine, coded in C++ and interfaced with Python.

Fig. 6 shows how the XML configuration files are used as input and output of the C++ library (which has a Python interface) to simulate the dynamic evolution of the crowd. The contents of each XML configuration file are detailed below. All units are expressed in the International System (SI). One example of each file (used for the practical case example presented in Sec. 4.3) is provided in App. C. We begin with the `Parameters.xml` file:

- **Parameters**:
    - ⋆ Directory in which STATIC files are stored,
    - ⋆ Directory in which DYNAMIC files are stored,
    - ⋆ Time step `TimeStepMechanical` of the Velocity-Verlet algorithm used to solve the dynamics,
    - ⋆ Duration `TimeStep` of one decisional loop, after which $\mathbf{F}_p$ and $\tau_p$ can change, typically a fraction of a second.

"STATIC" files contain information that does not change throughout a simulation, namely:

- **Geometry**:
    - ⋆ `Dimensions` of the simulation area,
    - ⋆ List of 'obstacles' (notably, `wall`), defined as ordered lists of `corners` (i.e., the vertices that are connected by the zero-width wall faces). Each obstacle is made of a given material, whose ID must be specified.
- **Materials** (for obstacles as well as agents):
    - ⋆ *Intrinsic* **properties**: Young's modulus $E$, shear modulus $G$ for 2D materials relative to a unique material ID.
      Note: these moduli enter the stiffness of the springs that are used to model contact interactions, following common practice in the discrete-element method

244     (the formulae are derived from [45, 46], also see Fig. 5)

$$k^{\perp} = \left( \frac{4G_1 - E_1}{4G_1^2} + \frac{4G_2 - E_2}{4G_2^2} \right)^{-1}, \tag{5}$$

$$k^{\parallel} = \left( \frac{6G_1 - E_1}{8G_1^2} + \frac{6G_2 - E_2}{8G_2^2} \right)^{-1}, \tag{6}$$

245     ⋆ **Binary physical properties** that are not reducible to intrinsic ones: damping
246       coefficient $\gamma^{\perp}$ perpendicular to the contact surface, damping coefficient $\gamma^{\parallel}$
247       tangential to the contact surface, dynamic friction coefficient during slip $\mu^{\text{dyn}}$.
248       These need to be defined for all pairs of materials.
249   • **Agents**:
250     ⋆ ID of the agent,
251     ⋆ Mass,
252     ⋆ Height,
253     ⋆ Moment of inertia,
254     ⋆ Inverse timescale for translational friction $1/t^{(\text{translational})}$ (`FloorDamping`),
255     ⋆ Inverse timescale for rotational damping $1/t^{(\text{rotational})}$ (`AngularDamping`),
256     ⋆ Constitutive shapes (5 for a pedestrian):
257       ▷ ID of the constitutive material,
258       ▷ Radius,
259       ▷ Initial `position` relative to agent's centre of mass.
260     Note: the composite shapes order is important, because the body's orientation will
261     be determined based on the first and last composite shapes. For a pedestrian, the
262     first composite shape should be the left shoulder, and the last one should be the
263     right shoulder.
264   "DYNAMIC FILES" are used both as input and output of the C++ library, and they contain
265   information that changes during the execution of the code, namely:
266   • **AgentDynamics** (current state of the agents):
267     ⋆ **Kinematic** quantities for each agent:
268       ▷ Position **r** of the center of mass,
269       ▷ Velocity **v** of the center of mass,
270       ▷ Orientation (`Theta`) of the body concerning the x-axis, that is, the angle $\theta$
271         between the gaze of the agent when looking straight ahead, and the x-axis
272         (see Fig. 11c),
273       ▷ Angular velocity (`Omega`).
274     ⋆ **Dynamic** quantities for each agent (not written in the output files):
275       ▷ Propulsion force $\mathbf{F}_{\text{p}}$ (`Fp`),
276       ▷ Driving torque for the torso $\tau_{\text{p}}$ (`Mp`).
277   Note: all angular quantities are given relative to the z-axis, with the trigonometric
278   convention.
279   • **AgentInteractions**:
280     ⋆ Normal force $\mathbf{F}_{s^{(j)} \to s^{(i)}}^{\perp \text{contact}}$ (`Fn`), tangential force $\mathbf{F}_{s^{(j)} \to s^{(i)}}^{\parallel \text{contact}}$ (`Ft`), and tangential spring
281       elongation (`TangentialRelativeDisplacement`) also known as slip (see
282       Sec. A.1.1) between all pairs of composite shapes in contact (that do not belong
283       to the same agent),
284     ⋆ Normal force $\mathbf{F}_{w \to s^{(i)}}^{\perp \text{contact}}$, tangential force $\mathbf{F}_{w \to s^{(i)}}^{\parallel \text{contact}}$, and slip between all pairs (wall –
285       composite shape) in contact.
286   Note 1: using the symmetries of forces and spring elongation, we only list the values
287   once for each pair (composite shape – composite shape or wall – composite shape) in

288     contact.

289     Note 2: this file is only provided if there are contacts between agents or between agents
290     and walls. No such file is needed in the initial configuration, provided there are no
291     overlaps; the output file can be used *unchanged* for the next run.

## 3.2   Mechanical layer

293 In Fig. 6, the mechanical layer *CrowdMechanics* is a C++ shared library that handles the
294 dynamics of the agents described in Sec. 2.2. Calling instructions from C++ and Python are
295 provided in the online tutorials [32].

## 3.3   Python classes

297 The Python wrapper mirrors the foregoing structure, insofar as it contains Python classes
298 corresponding to the foregoing XML configuration files. However, since it also allows
299 generating a synthetic crowd based on anthropometric statistics and visualising it in 2D and
300 in 3D, additional Python classes needed to be defined. The following classes and 'dataclasses'
301 (which contain the statistics and measurements relevant for the generation of the crowd or
302 the agent) are provided:

303     ⋆ *Crowd* **class:** group of *Agent* objects.
304     The class contains methods to generate a crowd that abides by the measurement
305     constraints of *CrowdMeasures* and to position the agents either on a grid or using the
306     packing algorithm detailed in App. D.

307     ⋆ *CrowdMeasures* **dataclass**: Collection of dictionaries representing the characteristics.
308     By default, it contains ANSURII-based anthropometric statistics. But the user can
309     define custom normal distributions for each agent's attributes (e.g., pedestrian
310     bideltoid breadth, bike top tube length).

311     ⋆ *Agent* **class**: represents a single pedestrian (or bike rider, etc.)

312     ⋆ *AgentMeasures* **dataclass**: Collection of attribute measurements (e.g., chest depth,
313     mass, height for pedestrians; handlebar length, total bike weight for bikes). The
314     attribute values are taken from *CrowdMeasures* if the agent is instantiated from a
315     *Crowd*; alternatively, they can be specified manually if agents are created one by one.

316     ⋆ *InitialPedestrian* **class**: 2D and 3D contour shapes of a reference pedestrian.
317     The 2D shape consists of 5 overlapping discs, whose outer contour matches that of a
318     cryogenic specimen at shoulder's height[5] (see Sec. 2.1 and Fig. 3). There is one 2D
319     template that applies to both men and women, and separate 3D templates: one for
320     men and one for women. To further emphasise the versatility of the file structure, an
321     *InitialBike* **class** was also defined to represent the shape of a rider on a bike, that is to
322     say, a top-down approximate orthogonal projection of the bike and the rider.
323     The 3D shape takes the form of a dictionary where each key corresponds to a specific
324     altitude and each value is a `Shapely.MultiPolygon` object. Each
325     `Shapely.MultiPolygon` contains multiple `Shapely.Polygon` objects, each of
326     which is a polygon representing a distinct part of the body, such as a finger, an arm or a
327     leg, etc. This is illustrated in Fig. 7.

328     ⋆ *Shapes2D* **class**: 2D shape of a particular agent (pedestrian, bike, ...).
329     The 2D pedestrian's shape is obtained by transforming the reference 2D shape
330     (*InitialPedestrian* **class**) to match the measurements specified in *AgentMeasures*.
331     More precisely, the radii of the five disks are uniformly rescaled to match the specified

---

[5] More precisely, we consider the horizontal slice at the altitude used to measure bideltoid breadth in our 186.6 cm-tall reference cryogenic male specimen; the same altitude was used to measure the bideltoid breadth of the female specimen (whose feet are extended as if she were on tiptoe, resulting in an elongated posture).

chest depth, defined by the diameter of the middle disk. Additionally, a homothety centred at the agent's centroid is applied to the centres of each of the five composite disks to match the specified bideltoid breadth.

A similar process is applied for 2D bike shapes; it hinges on the application of homotheties to each composite shape of the reference bike.

* ⋆ **Shapes3D class**: 3D shape of a particular agent (only for pedestrians at present).

Starting from the reference pedestrian (**InitialPedestrian class**), the dimensions of each `Shapely.Polygon` at various altitudes are adjusted along with the altitude values themselves. Specifically, a vertical homothety is applied to ensure that the resulting shape matches the desired pedestrian height. Additionally, a homothety is applied to each contour of our reference cryogenic specimen defined in the *InitialPedestrian* class; the centre of the homothety is set at the mean of the x and y coordinates of each polygon's centroid. The scaling factors $s_{init}$ for the homothety are selected to match the chest depth and bideltoid breadth specified in *AgentMeasures*.

In detail, the chest depth is defined as the maximum distance between two points along the orientation-axis of the `Shapely.MultiPolygon` (i.e., the x-axis if the agent is turned to the right, corresponding to $\theta = 0$) in the slice at the torso's height (the altitude used to measure the bideltoid breadth of the cryogenically preserved specimen). The bideltoid breadth is defined as the maximum distance between two points along the axis orthogonal to the orientation-axis of the `Shapely.MultiPolygon` (the y-axis if $\theta = 0$) at the foregoing altitude.

To avoid inflating the head and feet because of the homothety, the scaling factors $\mathbf{s}_{init}$ are modulated with the altitude $z$; the final scaling factor is $\mathbf{s}_{new}(z) = f(z, \mathbf{s}_{init})$, where $f(z, \mathbf{s})$ is a smooth, door-shaped function equal to $\mathbf{1}$ for altitudes above the neck and below the knees (meaning no rescaling in those regions) and to $\mathbf{s}$ elsewhere. This approach ensures that, unlike the head, the belly and abdominal regions are duly inflated and reflect morphological differences, particularly for bigger individuals. We are aware that more sophisticated statistical shape models [47] can infer a 3D body shape from a limited set of measurements; however, we chose not to use them to simplify the model as much as possible.

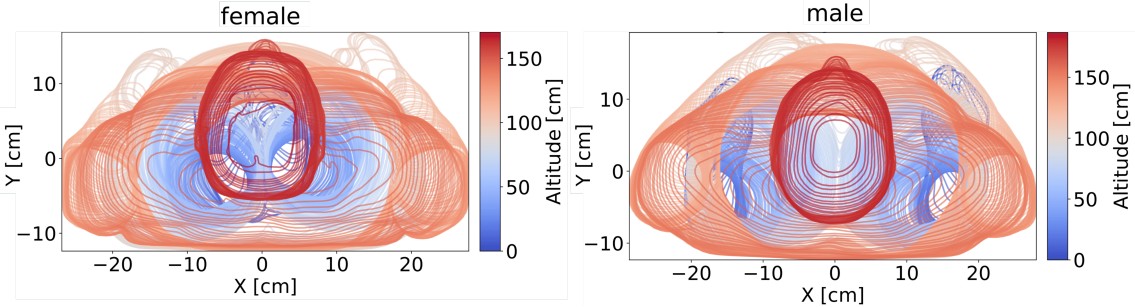

Figure 7: Superimposed cross-sectional contours of two cryogenically preserved bodies, sampled at 0.5 cm intervals. The body on the **left** is female, and the one on the **right** is male. Contours are extracted from the images of each section in [34]; upper-body regions appear reddish and lower-body regions bluish.

## 4  Discussion

### 4.1  Relevance of the use of 2D projections of standing pedestrians

In line with the dominant approach in pedestrian dynamics, our code primarily operates on 2D shapes, although it provides access to 3D visualisation. This simplification may be questioned because people have different heights, so that it may be inadequate to assess their contacts based on 2D projections at torso height. Shorter individuals (e.g., children, women) may have their heads at the level of the chests or shoulders of taller ones. Consequently, 2D crowd representations can vary significantly depending on the pedestrian's viewpoint; the practical impact of this perspective remains unclear. Our platform enables us to gauge the extent to which 2D projections reflect the packing conditions in a 3D crowd composed of adults of diverse heights. For this purpose, we generate a static 3D synthetic crowd based on the ANSURII database. In this example, pedestrian heights range from 155 cm to 178 cm for females and from 163 cm to 201 cm for males, with a mean height of 170 cm. As shown in Fig. 8, we present a comparison between the 2D projection of the scene–constructed from our pedestrian shape models–with cross-sections extracted from the corresponding 3D crowd at three distinct altitudes: the torso height of the smallest agent, the torso height of the tallest agent, and the mean torso height across the group. These comparisons reveal that perceived density can vary considerably with the pedestrian's height. Notably, the area covered at the mean torso height is closely matched by our 2D projection approach.

**Effective integration of leaning effects and hand contacts.** Our algorithm operates in 2D. As a consequence, it cannot *directly* integrate some previously evinced effects that may take place when densely packed people are destabilised, such as hand contacts or the push-induced forward leaning that may amplify pushing forces [48]. However, we would like to mention that they can be implemented *in an effective way* by amending the propulsion force $\mathbf{F}_p$ entering Eq. (1). Indeed, if a postural model (such as that proposed in [48]) can predict how a push propagates through a pedestrian (depending on their physical attributes and how they were pushed), then this agent's propulsion force $\mathbf{F}_p$ can be supplemented with this pushing force. Within LEMONS, one may also consider defining effective shapes corresponding to the situation of stretched arms and extended hands. Nonetheless, as we will see in the practical case study of Sec. 4.3, these refinements may be superfluous to describe the propagation of a push on flat ground.

Along similar lines, since pedestrians may stand and pivot on only their right or left leg, whereas the model computes torques along the body's central vertical axis, in some circumstances it might be necessary to account for the difference between the right or left leg's axis and the central axis. In principle, if the stepping dynamics are known, this can be achieved via the parallel-axis theorem, even though in practice it may prove complicated.

### 4.2  Mechanical tests

Tests for agent generation comprehensively cover all essential functions, including rotation operations, backup file handling, and file downloading. They also verify that the statistical properties of the generated agents, such as mean bideltoid breadth, mean chest depth, and standard deviation, accurately match the intended crowd statistics. To execute these tests, run from the root directory:

```
uv run pytest tests/configuration
```

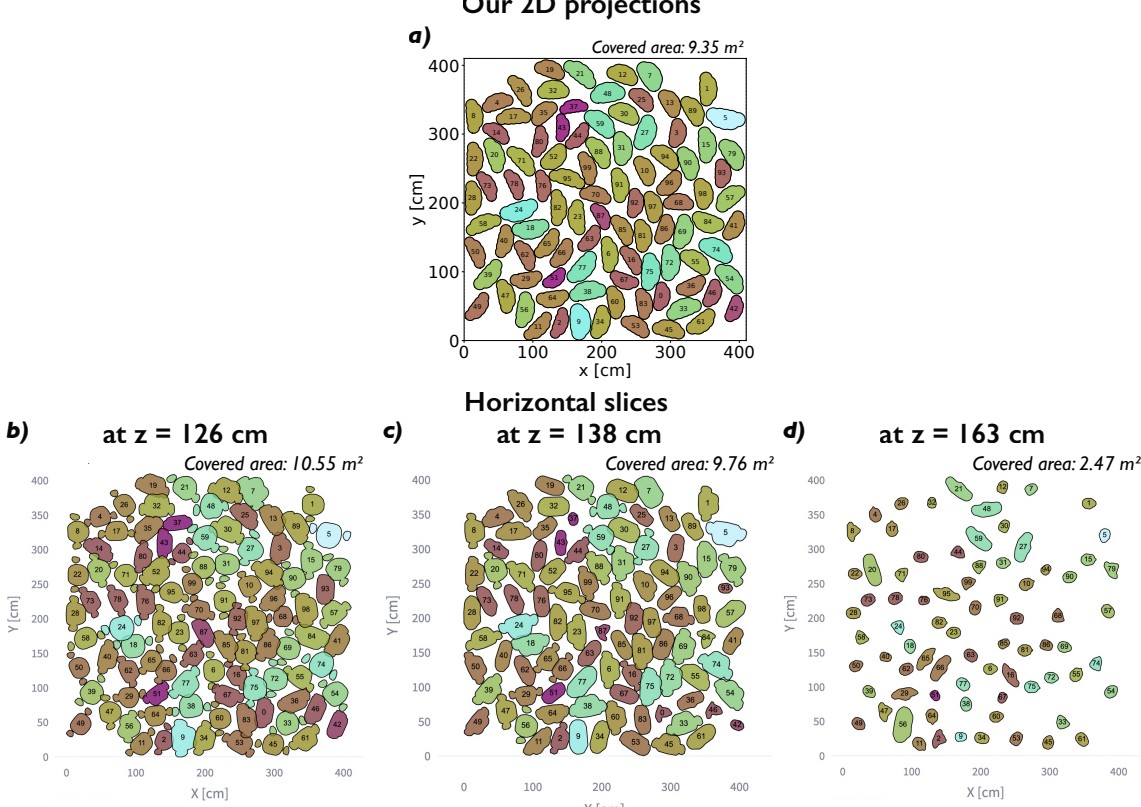

Figure 8: 2D representations of a pedestrian crowd at a density of about $6\,\text{ped}/\text{m}^2$. Panel **(a)** shows the 2D projections used in the simulations, while panels **(b)**, **(c)**, and **(d)** display horizontal cross-section of the 3D crowd at $z = 126\,\text{cm}$, $z = 138\,\text{cm}$, and $z = 163\,\text{cm}$, corresponding to the torso altitudes of the shortest pedestrian, the mean torso altitude, and the tallest pedestrian, respectively. Indicated areas are the sums of the shapes' areas without overlap correction.

Regarding the simulation engine, eight test suites covering distinct scenarios have been defined. These tests are designed to verify the behaviour of each mathematical term in the mechanical model; they are not intended as comparisons with experimental data. They rely on tolerance thresholds (detailed in the API documentation) that you can adjust if necessary. They should be repeated after each modification of the C++ code or data files and are also included in the continuous integration pipeline (see App. E). They all can be executed locally from the project root as follows:

1. Navigate to the `tests/mechanical_layer` directory.
2. Run the following command in your terminal:

```
./run_mechanical_tests.sh
```

The results of the eight test suites will appear directly within the Terminal.

3. If you further want to visualise the results of the tests as videos, run the following command in your terminal:

```
./make_tests_videos.sh
```

The script first prompts you for the path to your FFmpeg executable, which is required to generate movies from the simulation files. All videos are saved in the `tests/mechanical_layer/movies` directory. Once generated, you can review them and verify that they meet your expectations.

The eight test scenarios are as follows:

* **Agent pushing another agent** (`test_push_agent_agent` folder)
  Tests the force orthogonal to the contact surface, representing a damped spring interaction between two agents.
* **Agent colliding with a wall** (`test_push_agent_wall` folder)
  Tests the force orthogonal to the contact surface, representing a damped spring interaction between an agent and a wall.
* **Agent sliding over other agents** (`test_slip_agent_agent` folder)
  Tests the Coulomb friction interaction between two agents as one slides over the other.
* **Agent sliding over a wall** (`test_slip_agent_wall` folder)
  Tests the Coulomb friction interaction between an agent and a wall as the agent slides along it.
* **Agent translating and relaxing** (`test_t_translation` folder)
  Tests the behaviour as an agent undergoes a translation and gradually relaxes to a stationary state (no motion), due to the fluid-like force with the damping coefficient of $1/t^{\text{(translation)}}$.
* **Agent rotating and relaxing** (`test_t_rotation` folder)
  Tests the behaviour as an agent rotates and gradually relaxes to a stationary state (no motion), due to the fluid-like torque with the damping coefficient of $1/t^{\text{(rotation)}}$.
* **Agent rolling over other agents without sliding**
  (`test_tangential_spring_agent_agent` folder)
  Tests the force tangential to the contact surface, representing a damped spring interaction between two agents.
* **Agent rolling over a wall without sliding** (`test_tangential_spring_agent_wall` folder)
  Tests the force tangential to the contact surface, representing a damped spring interaction between an agent and a wall.

These tests yielded outcomes that concord with the expectations for the implemented mechanical model (the videos are provided in the supplemental material [33]).

## 4.3  Practical case study

We will detail here how to perform a simulation of a push that propagates through a queue of closely standing people, a scenario that mirrors recent experiments by Feldmann et al. [36] (and by Wang and Weng [48]). Illustrative snapshots[6] of the experiments and simulations are shown in Fig. 9. For us, the interest of this scenario is that it largely relies on the *mechanical* modelling layer, and only a little on the *decisional* layer (which we recall the user of our *mechanical* code can choose freely). Consistently with the scope of this *Codebase* paper, we defer the details of the experimental comparison to another publication.

**Estimation of the mechanical parameters.**

The parameters employed in the simulations are detailed in Tab. 1. They were selected to align with the experimental results of [36] and to produce credible output in simple scenarios. We start by justifying the consistency of the estimates for the key mechanical parameters.

---

[6]Pedestrian size corresponds to the area (in m$^2$) of the 2D shapes used in the simulations. Because chest depth and bideltoid breadth were not measured during the experiment (but the mass and height were), these dimensions were obtained from the ANSURII anthropometric database by selecting, for each participant, the values corresponding to their recorded body mass and height. This procedure yields individualised 2D shapes consistent at least qualitatively with the observed body proportions in the videos.

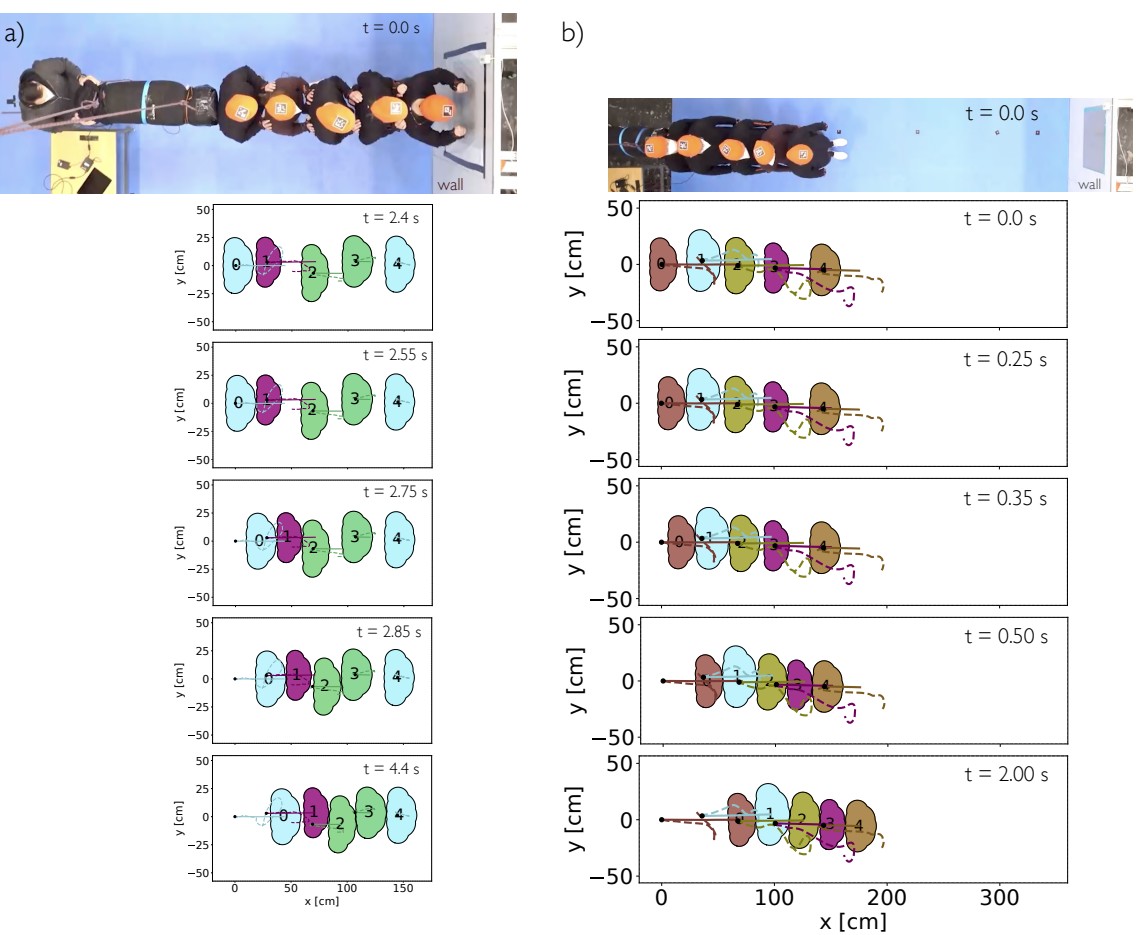

Figure 9: Simulated trajectories for two pushing scenarios based on Feldmann et al. [12]. Dashed lines show the *measured* head trajectories, while solid lines show *model predictions* with fine-tuned parameters, which are kept identical across the two scenarios. **(a)** Near-wall configuration with arms held in front. **(b)** Far-from-wall configuration with arms initially alongside the body and later raised for protection.

The translation relaxation time $t^{(\text{transl})}$ should be around the duration of one step, i.e., 0.5 s. More precisely, Li and colleagues [49] found that, when one suddenly pushes a static pedestrian, they come to a halt after a distance that grows as $a \simeq 0.02$ times the impulse $\mathcal{I}$, i.e., the time-integral of the pushing force. Supposing that this force is constant over $\Delta t$ and then vanishes, integrating Eq. (1) (with motion constraint along a single dimension) in the absence of any other propulsion force yields a scalar speed

$$v(t) = \frac{t^{(\text{transl})}\mathcal{I}}{m\Delta t}\left[\exp\left(-\frac{\max(0, t - \Delta t)}{t^{(\text{transl})}}\right) - \exp\left(-\frac{t}{t^{(\text{transl})}}\right)\right], \tag{7}$$

hence a halting distance

$$\Delta y = \int_0^{+\infty} v(t)\,\mathrm{d}t = \frac{t^{(\text{transl})}\mathcal{I}}{m}, \tag{8}$$

whence taking $m = 53$ kg we arrive at $t^{(\text{transl})} \approx 1$ s, larger but comparable to our estimate of 0.5 s. The rotational relaxation time $t^{(\text{rot})}$ was taken equal to $t^{(\text{transl})}$, because both relaxations have a similar origin, namely the contact of the feet with the ground.

Turning to the Young's modulus $E_{\text{body}}$ (noting that assuming it to be uniform is a clear oversimplification, given the layered heterogeneity of the human body), we base our value on

reported measurements of the elastic modulus of sternal trabecular bone, located at the centre of the thorax [50]. These data give $E_{\text{body}} \approx 4.0 \times 10^7$ kg/(m s$^2$), which, after conversion from 3D to 2D by multiplying by a characteristic load length of 10 cm, supports our estimate in Tab. 1. The Young and shear moduli assigned to the walls were taken from values for concrete and were converted from 3D to 2D using the same characteristic load length.

The coefficient of sliding friction $\mu_{\text{body}}^{\text{dyn}} = 0.4$ lies in the typical range of values for dry irregular surfaces (e.g., leather on oak) [51].

The damping coefficient for the direction orthogonal to the pedestrian–wall contact surface is set to the value reported by [41] for thoracic extension in wood–bones impact, accounting for energy dissipation due to air in the lungs and blood in the thoracic vessels being displaced during impact. To further refine these estimates, additional experiments using clothed, unembalmed cadavers could be conducted, following those already performed on granular materials, such as for wood in [52].

| Parameter | Description | Value | Compatible with |
|---|---|---|---|
| $t^{\text{(transl)}}$ | Relaxation time for translational motion | 0.22 s | [49] |
| $t^{\text{(rot)}}$ | Relaxation time for rotational motion | 0.22 s | |
| $E_{\text{body}}$ | Young modulus for the body (human naked material) | 4.0e6 kg s$^2$ | [53] |
| $G_{\text{body}}$ | Shear modulus for the body (human naked material) | 1.38e6 kg/s$^2$ | [53] |
| $\gamma_{\text{body}}^{\perp}$ | Damping for pedestrian-pedestrian contact in the direction orthogonal to the surface contact | 0.7e3 kg/s | |
| $\gamma_{\text{body}}^{\parallel}$ | Damping for pedestrian-pedestrian contact in the direction parallel to the surface contact | 0.7e3 kg/s | |
| $\mu_{\text{body}}^{\text{dyn}}$ | Kinetic friction for pedestrian-pedestrian contact | 0.4 | [51] |
| $E_{\text{wall}}$ | Young modulus for the wall (concrete) | 1.7e9 kg/s$^2$ | [54] |
| $G_{\text{wall}}$ | Shear modulus for the wall (concrete) | 7.1e8 kg/s$^2$ | [54] |
| $\gamma_{\text{wall}}^{\perp}$ | Damping for pedestrian-wall contact in the direction orthogonal to the surface contact | 1.23e3 kg/s | [41] |
| $\gamma_{\text{wall}}^{\parallel}$ | Damping for pedestrian-wall contact in the direction parallel to the surface contact | 1.23e3 kg/s | |
| $\mu_{\text{wall}}^{\text{dyn}}$ | Kinetic friction for pedestrian-wall contact | 0.5 | [51] |

Table 1: Parameter values used in the practical case study. The elastic moduli are expressed for 2D systems. We multiplied the measured 3D moduli by a characteristic load length of 0.1 m to convert from kg/(m s$^2$) to the 2D units kg/ s$^2$.

**Set the working environment and generate the desired configuration files**

We now describe how to run the code for this practical case study, focusing on the scenario shown in panel (a) of Fig. 9. Start by creating your desired crowd using the online platform, for example, eight pedestrians with anthropometric characteristics from the ANSURII database, arranged in a tightly packed configuration. Download the resulting configuration files to your local system. For instance, you can create a new directory called `Trial_1` and navigate into it. Create and configure a `Parameters.xml` file in this directory. Within `Trial_1`, add two subdirectories named `static` and `dynamic`. Place the configuration files obtained from the online platform into their respective folders. For reference, an example of the recommended directory structure is shown below:

```
.
|-- Parameters.xml
|-- static/
|    |-- Agents.xml
|    |-- Geometry.xml
|    |-- Materials.xml
```

```
513   |-- dynamic/
514   |     |-- AgentDynamics.xml
515
```

Finally, modify the `Geometry.xml` file to define the desired geometry, and adjust the `AgentDynamics.xml` file to set the appropriate initial propulsion force and torque. Refer to App. C for the configuration files used in this practical case.

**Run the simulation**

First of all, you need to navigate to the root of the `src/mechanical_layer` directory and build the project:

```
cmake -H. -Bbuild -DBUILD_SHARED_LIBS=ON
cmake --build build
```

Run the Python code provided below, making any necessary modifications to suit your needs. The simulation results will be saved automatically in the `outputXML/` directory. Each output file follows the naming pattern `AgentDynamics output t=TIME_VALUE.xml`, where `TIME_VALUE` indicates the corresponding simulation time or a unique identifier for that run. First, we import the recorded external force data corresponding to the initial push applied to the leftmost agent in the row, and then construct an interpolation function so that it can be used as the propulsion-force input for the mechanical layer (all other agents have a self-propulsion force equal to **0**):

```python
import xml.etree.ElementTree as ET
from pathlib import Path

import pandas as pd
from scipy.interpolate import interp1d

# === Import of external force data ===
dataPath = Path("../../../../data/tutorial_mechanical_layer/push_Feldmann/Wed_03_m_wiW_row4_14_w_s_b_p_n_u")
df = pd.read_csv(
    dataPath / "external_force_per_mass.txt",
    sep=r"\s+",
    header=0,
    names=["time [s]", "force per mass [N/m]"],
)

# === Read mass of agent with Id 0 from XML configuration ===
XMLtree = ET.parse("static/Agents.xml")
agentsTree = XMLtree.getroot()

mass_agent_0 = 0.0
for agent in agentsTree:
    if int(agent.attrib["Id"]) == 0:
        mass_agent_0 = float(agent.attrib["Mass"])
        break

print(f"Mass of agent 0: {mass_agent_0} kg\n")

# === Build interpolator for the external force on agent 0 ===
_push_agent0_interp = interp1d(
    df["time [s]"].values,
    df["force per mass [N/m]"].values * mass_agent_0,  # Multiply force per unit mass by mass to obtain the total force
        ↪ on agent 0.
    kind="linear",
    fill_value=0.0,  # zero force outside the sampled time range
)
```

Now you can run the mechanical layer:

```python
import ctypes
from pathlib import Path
import numpy as np
from shutil import copyfile
import xml.etree.ElementTree as ET

# === Simulation Parameters ===
dt = 0.1  # Time step for the decisional layer (matches "TimeStep" in Parameters.xml)
Ndt = 100  # How many dt will be performed in total

# === Paths Setup ===
outputPath = Path("outputXML/")  # Directory to store output XML files
inputPath = Path("inputXML/")  # Directory to store input XML files
outputPath.mkdir(parents=True, exist_ok=True)  # Create directories if they don't exist
inputPath.mkdir(parents=True, exist_ok=True)
```

```
589    # === Loading the External Mechanics Library ===
590    # Adjust filename for OS (.so for Linux, .dylib for macOS)
591    Clibrary = ctypes.CDLL("../../src/mechanical_layer/build/libCrowdMechanics.dylib")
592
593
594    agentDynamicsFilename = "AgentDynamics.xml"
595
596    # Prepare the list of XML files that will be passed to the DLL/shared library
597    files = [
598        b"Parameters.xml",
599        b"Materials.xml",
600        b"Geometry.xml",
601        b"Agents.xml",
602        agentDynamicsFilename.encode("ascii"),  # Convert filename to bytes (required by ctypes)
603    ]
604    nFiles = len(files)  # Number of configuration files to be passed
605    filesInput = (ctypes.c_char_p * nFiles)()  # Create a ctypes array of string pointers
606    filesInput[:] = files  # Populate array with the XML file names
607
608    # === Main Simulation Loop ===
609    for t in range(Ndt):
610        print("Looping the Crowd mechanics engine - t=%.1fs..." % (t * dt))
611
612        # 1. Save the current AgentDynamics file as input for this step (can be used for analysis later)
613        copyfile("dynamic/" + agentDynamicsFilename, str(inputPath) + rf"/AgentDynamics input t={t * dt:.1f}.xml")
614
615        # 2. Call the external mechanics engine, passing in the list of required XML files
616        Clibrary.CrowdMechanics(filesInput)
617
618        # 3. Save the updated AgentDynamics output to results folder (can be used for analysis later)
619        copyfile("dynamic/" + agentDynamicsFilename, str(outputPath) + rf"/AgentDynamics output t={(t + 1) * dt:.1f}.xml")
620
621        # 4. If the simulation produced an AgentInteractions.xml file, save that as well (optional output)
622        try:
623            copyfile("dynamic/AgentInteractions.xml", str(outputPath) + rf"/AgentInteractions t={(t + 1) * dt:.1f}.xml")
624        except FileNotFoundError:
625            # If the AgentInteractions file does not exist, skip copying
626            pass
627
628        # === Decision/Controller Layer for Next Step ===
629        # Read the output AgentDynamics XML as input for the next run.
630        # This is where you (or another program) can set new forces/moments for each agent for the next simulation step.
631        XMLtree = ET.parse("dynamic/" + agentDynamicsFilename)
632        agentsTree = XMLtree.getroot()
633
634        # -- Assign random forces/moments to each agent --
635        for agent in agentsTree:
636            # Create new <Dynamics> tag for the agent (as the output file doesn't have it)
637            dynamicsItem = ET.SubElement(agent, "Dynamics")
638
639            # Assign random force, and random moment
640            dynamicsItem.attrib["Fp"] = f"{np.random.normal(loc=200, scale=200):.2f},{np.random.normal(loc=0, scale=50):.2f}
641    ↪ "
642            dynamicsItem.attrib["Mp"] = f"{np.random.normal(loc=0, scale=5):.2f}"
643
644        # Write the modified XML back, to be used in the next iteration
645        XMLtree.write("dynamic/" + agentDynamicsFilename)
646        # ================================================

647
648    # After all simulation steps are complete, print a final message.
649    print(f"Loop terminated at t={Ndt * dt:.1f}s!")
```

### Generate plots and create a video from output files

A plot of the scene can be generated from each input/output file under PNG format using the
Python wrapper. To begin, you need to install the required Python packages, which you can
quickly do by setting up a virtual environment using uv as follows (from the root directory of
the project):

```
python -m pip install --upgrade pip
pip install uv
uv sync
```

You can then run the following Python script within your working environment:

```
import matplotlib.pyplot as plt

import configuration.backup.dict_to_xml_and_reverse as fun_xml  # For converting XML to dictionary and vice versa
from configuration.models.crowd import create_agents_from_dynamic_static_geometry_parameters  # For creating agents
        ↪ based on XML data
from streamlit_app.plot import plot  # For plotting crowd data

# === Prepare the folders ===
```

```
673    # Define the paths to the folders you'll use
674    outputPath = Path("outputXML")
675    staticPath = Path("static")
676    plotsPath = Path("plots")
677    plotsPath.mkdir(parents=True, exist_ok=True)  # Create plots directory if it doesn't exist
678
679    # Remove any old '.png' files in the plots directory
680    for file in plotsPath.glob("*.png"):
681        os.remove(file)
682
683    # === Load static XML files ===
684    # Read the Agents.xml file as a string and convert it to a dictionary
685    with open(staticPath / "Agents.xml", encoding="utf-8") as f:
686        crowd_xml = f.read()
687    static_dict = fun_xml.static_xml_to_dict(crowd_xml)
688
689    # Read the Geometry.xml file as a string and convert it to a dictionary
690    with open(staticPath / "Geometry.xml", encoding="utf-8") as f:
691        geometry_xml = f.read()
692    geometry_dict = fun_xml.geometry_xml_to_dict(geometry_xml)
693
694    # === Loop over time steps ===
695    for t in range(Ndt):
696        current_time = (t + 1) * dt
697
698        # Check if the dynamics file exists; if not, skip to the next time step
699        dynamics_file = outputPath / f"AgentDynamics output t={current_time:.1f}.xml"
700        if not dynamics_file.exists():
701            print(f"Warning: {dynamics_file} not found, skipping.")
702            continue
703
704        # === Read and process the dynamics XML file ===
705        # Read the current dynamics XML file as a string and convert it to a dictionary
706        with open(dynamics_file, encoding="utf-8") as f:
707            dynamic_xml = f.read()
708        dynamic_dict = fun_xml.dynamic_xml_to_dict(dynamic_xml)
709
710        # Create a crowd object using the configuration files data
711        crowd = create_agents_from_dynamic_static_geometry_parameters(
712            static_dict=static_dict,
713            dynamic_dict=dynamic_dict,
714            geometry_dict=geometry_dict,
715        )
716
717        # Plot the crowd
718        plot.display_crowd2D(crowd)
719        plt.savefig(plotsPath / rf"crowd2D_t={t:d}.png", dpi=300, format="png")
720        plt.close()
```

Additionally, simulated and measured trajectories can be overlaid in each plot, as in Fig. 9 (which is detailed in the online tutorial [32]). The resulting series of PNG images can then be combined into a video using FFmpeg. Representative frames are shown in Fig. 9, and the complete video is provided in the supplemental materials [33].

## 4.4 Extension to arbitrary shapes

This software was designed to facilitate further development, particularly by including a wider variety of agents, e.g., people carrying a backpack, children, etc. To prove this point, we chose a very different type of shapes, namely, bicycles, and implemented them in the 2D agent generation on the online application [55]. To access this feature, navigate to the CROWD tab, then in the sidebar under DATABASE ORIGIN, select the Custom statistics option, and set the desired proportion of bicycles within the crowd. The bicycle agent has been simplified to two overlapping rectangular polygons: one representing the front and rear wheels, and the other representing the seated rider and handlebars. The statistics of the dimensions of these shapes are adjustable. Note, however, that the simulation code does not model the mechanical interactions with bicycles, which we consider less relevant and more complex than those between pedestrians. An example of such a heterogeneous crowd is shown in Fig. 10.

The configuration file synthesising the crowd can be downloaded in XML format; it is simpler than the configuration files for a pedestrian-only crowd. The file includes a list of agents, each containing the following information: type (either pedestrian or bike), Id (an integer), Moment of inertia (in kg·m$^2$), FloorDamping ($t^{(transl)}$), AngularDamping ($t^{(rot)}$), and Shapes. For agents of type bike, the Shapes tag contains two tags: bike (corresponding to the front and rear wheels) and rider (corresponding to the human on the bicycle and the handlebars). Within the bike tag, several other tags are included: type

(rectangle), `material` (iron, human clothes, etc.), `min_x`, `min_y`, `max_x`, and `max_y`, which transparently define the rectangle's boundaries in absolute coordinates. The `rider` tag follows a similar structure. For agents of type `pedestrian`, a similar structure is used. However, within the `Shapes` tag, there are sub-tags `disk0`, `disk1`, up to `disk4`, each of which specifies the following attributes: `type` (disk), `radius`, `material`, and `x`, `y` (the position of the disk's centre in absolute coordinates).

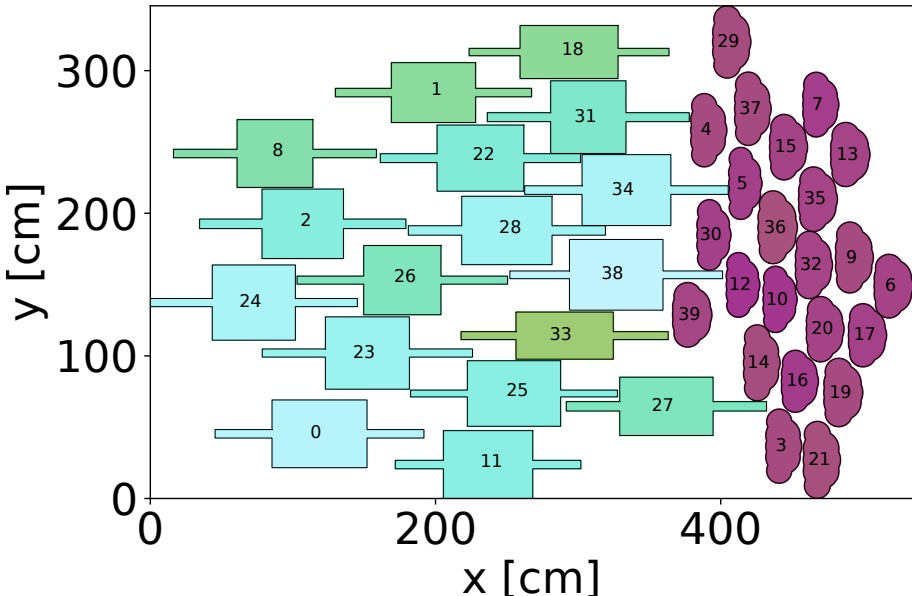

Figure 10: Heterogeneous crowd of 40 agents (17 bicycles + 23 pedestrians) with uniform orientation. Agent area is colour-coded using the Hawaii colourmap [56], from purple (smallest) to blue (largest). This example is not intended to be realistic, but rather to showcase that the pedestrian generation code can be easily generalised.

# 5 Conclusion

In summary, we have released an open-source numerical tool to help modellers simulate the dynamics of pedestrians in 2D and visualise the output in 2D and 3D. This tool is *not* a pedestrian simulation software (because the decisional components, notably the desired speeds and directions, should be given as input), but it adds a substantial contribution to the field, especially for the study of dense crowds, in that it promotes realistic 2D projections of pedestrians, grounded in anthropometric data and much more faithful than the typical circular assumption, and it computes contact forces derived from Physics. To make the code as broadly accessible to the public as possible, we have released an online platform for generating and visualising agents, a computationally efficient C++ library for dynamical simulations, and an easy-to-use Python wrapper to run all scripts.

To let the tool evolve with the field, a generic XML format for configuration files has been proposed. Currently, the tool can only generate bare or clothed adult men and women, as well as cyclists. However, thanks to the generic file format, other shapes may be included in the future, such as children, people carrying a backpack, and people pushing a pushchair. Further in the future, it may also become relevant to extend the mechanical computations of contact forces to 3D.

## 768 Acknowledgements

769 The authors are grateful to David RODNEY for sharing his materials science expertise, to
770 Gaël HUYNH and Mohcine CHRAIBI for their words of advice about code structuring and
771 development.

772 **Author contributions** O.D.: Conception, C++ and Python Coding, Testing, Writing –
773 original draft. M.S.: C++ and Python Coding, Testing, Writing – review and editing. A.N.:
774 Conception, Supervision, some Python Coding, Writing – review and editing.

775 **Funding information** This work was conducted in the frame of the following projects:
776 French-German research project MADRAS funded in France by the Agence Nationale de la
777 Recherche (grant number ANR-20-CE92-0033), and in Germany by the Deutsche
778 Forschungsgemeinschaft (grant number 446168800), French project MUTATIS funded by
779 Agence Nationale de la Recherche (grant number ANR-24-CE22-0918). This project has also
780 received financial support from the CNRS through the MITI interdisciplinary programs. The
781 authors are not aware of any competing interests.

782
783

## 784 A  Equation of motion

### 785 A.1  Mechanical interactions

786 Consider two pedestrians, $i$ and $j$, represented by sets of disks $s^{(i)}$ and $s^{(j)}$ respectively. Each
787 disk center $s^i$ of pedestrian $i$ is positioned relative to pedestrian $i$'s center of mass $G_i$ through
788 the displacement vector $\mathbf{\Delta}_{i \to s^{(i)}}$, which points toward $s^{(i)}$ (see Fig. 11a). The pedestrian's
789 orientation is defined by the normal vector to the line connecting their first and last disks (see
790 Fig. 11c). The CoM of pedestrian $i$ moves with a translational velocity $\mathbf{v}_i$, and the pedestrian
791 rotates with an angular velocity $\omega_i$.

#### 792 A.1.1  Forces acting on the pedestrian centre of mass

793 The motion of a pedestrian $i$ can be broken down into two components: the motion of its
794 Center of Mass (CoM) and rotational motion. The motion of the CoM is determined by applying
795 the fundamental principle of dynamics at that point. When the shape $s^{(i)}$ of pedestrian $i$ (with
796 radius $R_{s(i)}$ and position $\mathbf{r}_{s(i)}$) comes into contact with the shape $s^{(j)}$ of pedestrian $j$ (with radius
797 $R_{s(j)}$ and position $\mathbf{r}_{s(j)}$), as illustrated in Fig. 11a, pedestrian $i$ experiences the following forces
798 (analogous forces are applied in the case of contact with a wall, illustrated in Fig. 11b):

799     ⋆ A damped-spring force orthogonal to the surface contact denoted as $\mathbf{F}^{\perp\text{contact}}_{s^{(j)} \to s^{(i)}}$, split into
800        its spring part denoted as $\mathbf{F}^{\perp\text{contact}}_{\text{spring}, s^{(j)} \to s^{(i)}}$, linear with the interpenetration depth and a
801        damping part denoted as $\mathbf{F}^{\perp\text{contact}}_{\text{damping}, s^{(j)} \to s^{(i)}}$, that can be expressed as:

802          ▷ $\mathbf{F}^{\perp\text{contact}}_{\text{spring}, s^{(j)} \to s^{(i)}} = \begin{cases} k^{\perp}_{\text{body}} \, h_{s^{(i)}s^{(j)}} \, \mathbf{n}_{s^{(j)} \to s^{(i)}} & \text{if } h_{s^{(i)}s^{(j)}} = R_{s^{(i)}} + R_{s^{(j)}} - |\mathbf{r}_{s^{(j)} \to s^{(i)}}| > 0 \text{ (overlap)} \\ \mathbf{0} & \text{otherwise} \end{cases}$

803          ▷ $\mathbf{F}^{\perp\text{contact}}_{\text{damping}, s^{(j)} \to s^{(i)}} = \begin{cases} -\gamma^{\perp}_{\text{body}} \, \mathbf{v}^{\perp}_{ij} & \text{if } h_{s^{(i)}s^{(j)}} > 0 \text{ (i.e. an overlap occurs)} \\ \mathbf{0} & \text{otherwise} \end{cases}$

804        where $\mathbf{n}_{s^{(j)} \to s^{(i)}}$ denotes the unitary vector normal to the surface contact pointing towards
805        $s^{(i)}$, $\mathbf{v}^{\perp}_{ij}$ describes the relative velocity at the contact point $C$ along the direction normal

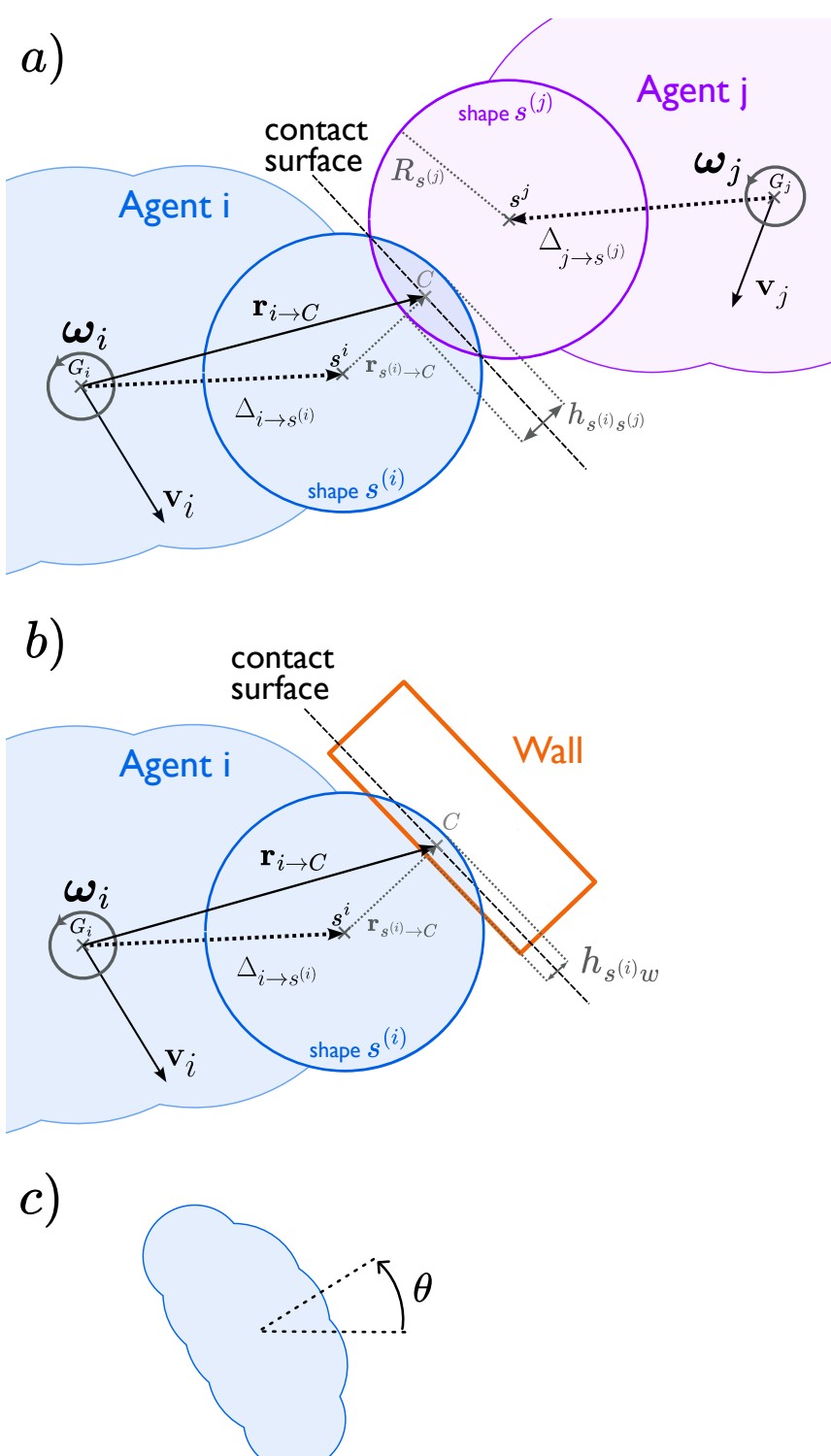

Figure 11: **(a)** Contact between two pedestrian bodies; **(b)** contact between a pedestrian body and a wall; **(c)** definition of pedestrian orientation. The contact surface is defined as the bisector of the shortest line segment connecting either the contours of two composite disks or the contour of a composite disk and a wall. The contact point $C$ is located at the midpoint of this segment.

to the surface contact and $\mathbf{r}_{s^{(j)} \to s^{(i)}}$ is the relative position of the two shapes in contact pointing towards shape $s^{(i)}$. $k_{\text{body}}^{\perp}$ represents the spring constant and $\gamma_{\text{body}}^{\perp}$ the damping

intensity in the normal direction for body-body contacts.

* A force, tangential to the contact surface that acts in the direction opposite to the slip. A straightforward way to model this force is through the Coulomb interaction to describe the stick and slip mechanism, and a damped spring to more precisely describe the stick phase. It can be written as:

$$\mathbf{F}^{\|\text{contact}}_{s^{(j)}\to s^{(i)}} = \begin{cases} k^{\|}_{\text{body}}\, \delta s \, \dfrac{-\mathbf{v}^{\|}_{ij}}{\left|\mathbf{v}^{\|}_{ij}\right|} - \gamma^{\|}_{\text{body}}\mathbf{v}^{\|}_{ij} & \text{if } k^{\|}_{\text{body}}\, \delta s + \gamma^{\|}_{\text{body}}\left|\mathbf{v}^{\|}_{ij}\right| < \mu^{\text{dyn}}_{\text{body}}\left|\mathbf{F}^{\perp\text{contact}}_{s^{(j)}\to s^{(i)}}\right| \ \ \text{(stick)} \\[2ex] \mu^{\text{dyn}}_{\text{body}}\left|\mathbf{F}^{\perp\text{contact}}_{s^{(j)}\to s^{(i)}}\right| \dfrac{-\mathbf{v}^{\|}_{ij}}{\left|\mathbf{v}^{\|}_{ij}\right|} & \text{otherwise} \ \ \text{(slip)} \end{cases} \tag{A.1}$$

where $\delta s$ represents the spring elongation and can be written as $\delta s = \left|\int_0^{\overset{\text{contact}}{\text{duration}}} \mathbf{v}^{\|}_{ij} dt\right|$

and $\mu^{\text{dyn}}_{\text{body}}$ denotes the dynamic friction coefficient. The force can be reshaped in a more condensed way as follows:

$$\mathbf{F}^{\|\text{contact}}_{s^{(j)}\to s^{(i)}} = \min\left( k^{\|}_{\text{body}}\, \delta s \, + \, \gamma^{\|}_{\text{body}}\left|\mathbf{v}^{\|}_{ij}\right| \, , \, \mu^{\text{dyn}}_{\text{body}}\left|\mathbf{F}^{\perp\text{contact}}_{s^{(j)}\to s^{(i)}}\right| \right) \dfrac{-\mathbf{v}^{\|}_{ij}}{\left|\mathbf{v}^{\|}_{ij}\right|} \tag{A.2}$$

* A self-propelling force $\mathbf{F}_p$, that converts decisions into actions;
* A fluid friction force, encompassing the effective backward friction with the ground over a simulation step cycle, controlled by the deformation of the body (biomechanical dissipation) expressed as $-m_i\,\mathbf{v}_i/t^{(\text{transl})}$, where $t^{(\text{transl})}$ is the characteristic relaxation time to the rest state.

### A.1.2 Torque for rotation of a pedestrian

The rotational motion of a pedestrian is obtained by applying the angular momentum theorem to the pedestrian's Center of Mass (CoM). This is done in its principal inertia base, projected along the z-axis (the out-of-plane axis). The pedestrian experiences torque due to the forces that are normal and tangential to the contact surface:

$$\tau_{G_i,\,s^{(j)}\to s^{(i)}} = \left\{ \mathbf{r}_{i\to C} \times \left( \mathbf{F}^{\|\text{contact}}_{s^{(j)}\to s^{(i)}} + \mathbf{F}^{\perp\text{contact}}_{s^{(j)}\to s^{(i)}} \right) \right\} \cdot \mathbf{u_z} \tag{A.3}$$

The self-propelling force and the fluid friction force act directly on the CoM, resulting in zero torque. To account for decision-making, a decisional torque $\tau_p$ is applied. Finally, analogous to the CoM equation, a fluid friction force accounting for floor contact and all mechanical dissipation mechanisms (including biomechanical effects) is incorporated as $-I_i\,\omega_i/t^{(\text{rot})}$. The computation of the moment of inertia $I_i$ is detailed in App. A.2.

### A.2 Moment of inertia calculation

Each pedestrian in our synthetic crowd is represented as a combination of five disks. While an analytical formula for the moment of inertia of such a configuration can be derived, it is quite cumbersome to write and implement numerically. Instead, we approximate the pedestrian's boundary using an $N$-sided polygon, defined by the set of vertices:

$$\{(x_1,y_1),(x_2,y_2),\dots,(x_{N+1},y_{N+1}): (x_1,y_1)=(x_{N+1},y_{N+1})\}, \tag{A.4}$$

where $(x_1,y_1)=(x_{N+1},y_{N+1})$ ensures the polygon is closed. Assuming pedestrian $i$'s mass $m_i$ is uniformly distributed within the polygon (yielding homogeneous mass density $\rho_i = m_i/\text{Polygon Area}$), the moment of inertia $I_i$ can be calculated via [57]:

$$I_i = \frac{\rho_i}{12}\sum_{j=1}^{N}\left(x_j y_{j+1} - x_{j+1}y_j\right)\left(x_j^2 + x_j x_{j+1} + x_{j+1}^2 + y_j^2 + y_j y_{j+1} + y_{j+1}^2\right). \tag{A.5}$$

### A.3   Mechanical equations summary

**Pedestrian CoM dynamics**

$$m_i \frac{\mathrm{d}\mathbf{v}_i}{\mathrm{d}t} = \mathbf{F}_p - m_i \frac{\mathbf{v}_i}{t^{(\text{transl})}} + \sum_{(s^{(j)}, s^{(i)}) \in C_i^{(\text{ped})}} \left( \mathbf{F}^{\|\text{contact}}_{s^{(j)} \to s^{(i)}} + \mathbf{F}^{\perp\text{contact}}_{s^{(j)} \to s^{(i)}} \right)$$
$$+ \sum_{(w, s^{(i)}) \in C_i^{(\text{wall})}} \left( \mathbf{F}^{\|\text{contact}}_{w \to s^{(i)}} + \mathbf{F}^{\perp\text{contact}}_{w \to s^{(i)}} \right) \tag{A.6}$$

**Interaction forces with a pedestrian**

$$\mathbf{F}^{\|\text{contact}}_{s^{(j)} \to s^{(i)}} = \min \left( k^{\|}_{\text{body}} \, \delta s \; + \; \gamma^{\|}_{\text{body}} \left| \mathbf{v}^{\|}_{ij} \right| \, , \; \mu^{\text{dyn}}_{\text{body}} \left| \mathbf{F}^{\perp\text{contact}}_{s^{(j)} \to s^{(i)}} \right| \right) \frac{-\mathbf{v}^{\|}_{ij}}{\left| \mathbf{v}^{\|}_{ij} \right|}$$

$$\mathbf{F}^{\perp\text{contact}}_{s^{(j)} \to s^{(i)}} = \mathbf{F}^{\perp\text{contact}}_{\text{spring}, s^{(j)} \to s^{(i)}} + \mathbf{F}^{\perp\text{contact}}_{\text{damping}, s^{(j)} \to s^{(i)}}$$

$$\mathbf{F}^{\perp\text{contact}}_{\text{spring}, s^{(j)} \to s^{(i)}} = \begin{cases} k^{\perp}_{\text{body}} \, h_{s^{(i)}s^{(j)}} \, \mathbf{n}_{s^{(j)} \to s^{(i)}} & \text{if } h_{s^{(i)}s^{(j)}} > 0 \text{ (i.e. an overlap occurs)} \\ \mathbf{0} & \text{otherwise} \end{cases} \tag{A.7}$$

$$\mathbf{F}^{\perp\text{contact}}_{\text{damping}, s^{(j)} \to s^{(i)}} = \begin{cases} -\gamma^{\perp}_{\text{body}} \, \mathbf{v}^{\perp}_{ij} & \text{if } h_{s^{(i)}s^{(j)}} > 0 \text{ (i.e. an overlap occurs)} \\ \mathbf{0} & \text{otherwise} \end{cases}$$

where

$$h_{s^{(i)}s^{(j)}} = R_{s^{(i)}} + R_{s^{(j)}} - \left| \mathbf{r}_{s^{(i)} \to s^{(j)}} \right|$$

$$\delta s = \left| \int_0^{\substack{\text{contact} \\ \text{duration}}} \mathbf{v}^{\|}_{ij} \, \mathrm{d}t \right|$$

$$\mathbf{v}^{\|}_{ij} = \mathbf{v}_{ij} - \mathbf{v}^{\perp}_{ij}$$

$$\mathbf{v}^{\perp}_{ij} = \left( \mathbf{v}_{ij} \cdot \mathbf{n}_{s^{(i)} \to s^{(j)}} \right) \mathbf{n}_{s^{(i)} \to s^{(j)}}$$

$$\mathbf{v}_{ij} = \mathbf{v}_{i,C} - \mathbf{v}_{j,C} \tag{A.8}$$

$$\mathbf{v}_{i,C} = \mathbf{v}_i + \omega_i \times \mathbf{r}_{i \to C}$$

$$\mathbf{r}_{i \to C} = \mathbf{\Delta}_{i \to s^{(i)}} + \mathbf{r}_{s^{(i)} \to C}$$

$$\mathbf{n}_{s^{(i)} \to s^{(j)}} = \frac{\mathbf{r}_{s^{(i)} \to s^{(j)}}}{\left| \mathbf{r}_{s^{(i)} \to s^{(j)}} \right|}$$

$$\mathbf{r}_{s^{(i)} \to s^{(j)}} = \mathbf{r}_j + \mathbf{\Delta}_{j \to s^{(j)}} - \left( \mathbf{r}_i + \mathbf{\Delta}_{i \to s^{(i)}} \right)$$

$$\mathbf{r}_{s^{(i)} \to C} = \left( R_{s^{(i)}} - \frac{h_{s^{(i)}s^{(j)}}}{2} \right) \mathbf{n}_{s^{(i)} \to s^{(j)}}$$

**Interaction forces with wall**

$$\mathbf{F}^{\|\text{contact}}_{w \to s^{(i)}} = \min \left( k^{\|}_{\text{wall}} \, \delta s_w \; + \; \gamma^{\|}_{\text{wall}} \left| \mathbf{v}^{\|}_{s^{(i)}w} \right| \, , \; \mu^{\text{dyn}}_{\text{wall}} \left| \mathbf{F}^{\perp\text{contact}}_{w \to s^{(i)}} \right| \right) \frac{-\mathbf{v}^{\|}_{iw}}{\left| \mathbf{v}^{\|}_{iw} \right|}$$

$$\mathbf{F}^{\perp\text{contact}}_{w \to s^{(i)}} = \mathbf{F}^{\perp\text{contact}}_{\text{spring}, w \to s^{(i)}} + \mathbf{F}^{\perp\text{contact}}_{\text{damping}, w \to s^{(i)}}$$

$$\mathbf{F}^{\perp\text{contact}}_{\text{spring}, w \to s^{(i)}} = \begin{cases} k^{\perp}_{\text{wall}} \, h_{s^{(i)}w} \, \mathbf{n}_{w \to s^{(i)}} & \text{if } h_{s^{(i)}w} > 0 \text{ (i.e. an overlap occurs)} \\ \mathbf{0} & \text{otherwise} \end{cases} \tag{A.9}$$

$$\mathbf{F}^{\perp\text{contact}}_{\text{damping}, w \to s^{(i)}} = \begin{cases} -\gamma^{\perp}_{\text{wall}} \, \mathbf{v}^{\perp}_{iw} & \text{if } h_{s^{(i)}w} > 0 \text{ (i.e. an overlap occurs)} \\ \mathbf{0} & \text{otherwise} \end{cases}$$

where

$$h_{s^{(i)}w} = R_{s^{(i)}} - |\mathbf{r}_{s^{(i)} \to w}|$$

$$\delta s_w = \left| \int_0^{\substack{\text{contact} \\ \text{duration}}} \mathbf{v}_{iw}^{\parallel} \, dt \right|$$

$$\mathbf{v}_{iw}^{\parallel} = \mathbf{v}_{i,C} - \mathbf{v}_{iw}^{\perp}$$

$$\mathbf{v}_{iw}^{\perp} = \left( \mathbf{v}_{i,C} \cdot \mathbf{n}_{s^{(i)} \to w} \right) \mathbf{n}_{s^{(i)} \to w}$$

$$\mathbf{v}_{i,C} = \mathbf{v}_i + \omega_i \times \mathbf{r}_{i \to C} \tag{A.10}$$

$$\mathbf{n}_{s^{(i)} \to w} = \frac{\mathbf{r}_{s^{(i)} \to w}}{|\mathbf{r}_{s^{(i)} \to w}|}$$

$$\mathbf{r}_{i \to C} = \boldsymbol{\Delta}_{i \to s^{(i)}} + \mathbf{r}_{s^{(i)} \to C}$$

$$\mathbf{r}_{s^{(i)} \to C} = \left( R_{s^{(i)}} - \frac{h_{s^{(i)}w}}{2} \right) \mathbf{n}_{s^{(i)} \to w}$$

$$\mathbf{r}_{s^{(i)} \to w} = \text{the vector from the center of } s^{(i)} \text{ to its nearest point on the wall } w$$

**Rotational dynamics**

$$I_i \frac{d\omega_i}{dt} = \tau_p - I_i \frac{\omega_i}{t^{(\text{rot})}} + \sum_{(s^{(j)}, s^{(i)}) \, \in \, \mathcal{C}_i^{(\text{ped})}} \tau_{G_i, s^{(j)} \to s^{(i)}} \\ + \sum_{(s^{(j)}, s^{(i)}) \, \in \, \mathcal{C}_i^{(\text{wall})}} \tau_{G_i, w \to s^{(i)}} \tag{A.11}$$

**Torques**

$$\tau_{G_i, s^{(j)} \to s^{(i)}} = \left\{ \mathbf{r}_{i \to C} \times \left( \mathbf{F}_{s^{(j)} \to s^{(i)}}^{\parallel \text{contact}} + \mathbf{F}_{s^{(j)} \to s^{(i)}}^{\perp \text{contact}} \right) \right\} \cdot \mathbf{u_z}$$

$$\tau_{G_i, w \to s^{(i)}} = \left\{ \mathbf{r}_{i \to C} \times \left( \mathbf{F}_{w \to s^{(i)}}^{\parallel \text{contact}} + \mathbf{F}_{w \to s^{(i)}}^{\perp \text{contact}} \right) \right\} \cdot \mathbf{u_z} \tag{A.12}$$

# B  Mechanical layer: agent shortlisting

To save computational power, the mechanical layer begins by identifying a subset of agents, dubbed the "mechanically active agents", for which a collision is likely/possible. The remaining agents are thereby considered as having no chance to collide with anything else during the execution of the code, and will therefore see their position evolve according to the "relaxation" part of equation (1) only. The shortlisting is performed in two steps:

(i) For each agent $i$, we establish a list of *neighbouring* agents and walls based on

- the radius $R_i$ of the agent – that is, the radius of the circle $\mathcal{C}_i$ centred on the agent's centre of mass, of which the agent's global shape is circumscribed;
- a global constant: the maximum – running – speed $v_{\text{max}} = 7 \, \text{m/s}$ of a pedestrian.

*Agent neighbours* of $i$ will be defined as agents $j$ for which the smallest distance between the borders of the circles $\mathcal{C}_i$ and $\mathcal{C}_j$ is smaller than the distance traveled by both agents at speed $v_{\text{max}}$ in a time `TimeStep` (ie twice the distance traveled at speed $v_{\text{max}}$ in a time `TimeStep`).

*Wall neighbours* of $i$ will be defined as walls for which the smallest distance between the border of the circle $\mathcal{C}_i$ and the wall is smaller than the distance travelled at speed $v_{\text{max}}$ in a time `TimeStep`).

(ii) We look at new positions of all agents after a uniform motion over time `TimeStep`, with velocity and angular velocity equal to

$$\nu^{(0)} \, \mathbf{e}^{(\text{target})} = \frac{\mathbf{F}_\text{p}}{m_i} \, \text{t}^{(\text{transl})} \quad \text{and} \quad \omega^{(0)} = \frac{\tau_\text{p}}{I_i} \, \text{t}^{(\text{rot})},$$

866   and check for overlaps with neighbours. In case of overlap with a *wall neighbour*, the
867   agent is considered "mechanically active", and in case of an overlap with an *agent*
868   *neighbour*, both agents are considered "mechanically active". Furthermore, at the end
869   of this process, we also add the *agent neighbours* of "mechanically active" agents.

870   Finally, agents with a significant difference between the three velocity components above
871   and the ones of their current state – i.e. above 1 cm/s, are added to the list.

## C   Configuration files example

**Parameters.xml file**

```
873
874  <?xml version="1.0" encoding="utf-8"?>
875  <Parameters>
876      <Directories Static="./static/" Dynamic="./dynamic/"/>
877      <Times TimeStep="0.05" TimeStepMechanical="2e-6"/>
878  </Parameters>
879
```

**Geometry.xml file**

```
880
881  <?xml version="1.0" encoding="utf-8"?>
882  <Geometry>
883      <Dimensions Lx="2.0526750" Ly="1.11766"/>
884      <Wall Id="0" MaterialId="concrete">
885          <Corner Coordinates="-0.2,-0.57395"/>
886          <Corner Coordinates="1.7526750,-0.57395"/>
887          <Corner Coordinates="1.7526750,0.543710"/>
888          <Corner Coordinates="-0.2,0.543710"/>
889          <Corner Coordinates="-0.2,-0.57395"/>
890      </Wall>
891  </Geometry>
```

**Materials.xml file**

```
893
894  <?xml version="1.0" encoding="utf-8"?>
895  <Materials>
896      <Intrinsic>
897          <Material Id="concrete" YoungModulus="1.70e+9" ShearModulus="7.10e+8"/>
898          <Material Id="human_clothes" YoungModulus="3.1e+06" ShearModulus="9e+05"/>
899          <Material Id="human_naked" YoungModulus="4.0e6" ShearModulus="1379310.3"/>
900      </Intrinsic>
901      <Binary>
902          <Contact Id1="concrete" Id2="concrete" GammaNormal="1.30e+03" GammaTangential="1.30e+03" KineticFriction="0.50"
903          ↪ />
904          <Contact Id1="concrete" Id2="human_clothes" GammaNormal="1.30e+03" GammaTangential="1.30e+03" KineticFriction="
905          ↪ 0.50"/>
906          <Contact Id1="concrete" Id2="human_naked" GammaNormal="1.23e+03" GammaTangential="1.23e+03" KineticFriction="
907          ↪ 0.50"/>
908          <Contact Id1="human_clothes" Id2="human_clothes" GammaNormal="1.30e+03" GammaTangential="1.30e+03"
909          ↪ KineticFriction="0.50"/>
910          <Contact Id1="human_clothes" Id2="human_naked" GammaNormal="1.30e+03" GammaTangential="1.30e+03" KineticFriction
911          ↪ ="0.50"/>
912          <Contact Id1="human_naked" Id2="human_naked" GammaNormal="0.7e3" GammaTangential="0.7e3" KineticFriction="0.4"/>
913      </Binary>
914  </Materials>
```

**Agents.xml file**

```
916
917  <?xml version="1.0" encoding="utf-8"?>
918  <Agents>
919      <Agent Type="pedestrian" Id="0" Mass="89.0" Height="1.794" MomentOfInertia="1.85" FloorDamping="4.50" AngularDamping
920      ↪ ="4.50">
921          <Shape Type="disk" Radius="0.09495" MaterialId="human_naked" Position="-0.015458,0.153544"/>
922          <Shape Type="disk" Radius="0.13058" MaterialId="human_naked" Position="0.008692,0.067374"/>
923          <Shape Type="disk" Radius="0.1365" MaterialId="human_naked" Position="0.013532,4e-06"/>
924          <Shape Type="disk" Radius="0.13058" MaterialId="human_naked" Position="0.008692,-0.067376"/>
925          <Shape Type="disk" Radius="0.09495" MaterialId="human_naked" Position="-0.015458,-0.153546"/>
926      </Agent>
927      <Agent Type="pedestrian" Id="1" Mass="63.0" Height="1.740" MomentOfInertia="1.02" FloorDamping="4.50" AngularDamping
928      ↪ ="4.50">
929          <Shape Type="disk" Radius="0.07826" MaterialId="human_naked" Position="-0.012738,0.144246"/>
930          <Shape Type="disk" Radius="0.10762" MaterialId="human_naked" Position="0.007162,0.063296"/>
```

```
931          <Shape Type="disk" Radius="0.1125" MaterialId="human_naked" Position="0.011152,-4e-06"/>
932          <Shape Type="disk" Radius="0.10762" MaterialId="human_naked" Position="0.007162,-0.063294"/>
933          <Shape Type="disk" Radius="0.07826" MaterialId="human_naked" Position="-0.012738,-0.144244"/>
934       </Agent>
935       <Agent Type="pedestrian" Id="2" Mass="86.0" Height="1.905" MomentOfInertia="1.78" FloorDamping="4.50" AngularDamping
936       ↪ ="4.50">
937          <Shape Type="disk" Radius="0.08591" MaterialId="human_naked" Position="-0.013986,0.168084"/>
938          <Shape Type="disk" Radius="0.11814" MaterialId="human_naked" Position="0.007864,0.073754"/>
939          <Shape Type="disk" Radius="0.1235" MaterialId="human_naked" Position="0.012244,4e-06"/>
940          <Shape Type="disk" Radius="0.11814" MaterialId="human_naked" Position="0.007864,-0.073756"/>
941          <Shape Type="disk" Radius="0.08591" MaterialId="human_naked" Position="-0.013986,-0.168086"/>
942       </Agent>
943       <Agent Type="pedestrian" Id="3" Mass="68.0" Height="1.902" MomentOfInertia="1.26" FloorDamping="4.50" AngularDamping
944       ↪ ="4.50">
945          <Shape Type="disk" Radius="0.09565" MaterialId="human_naked" Position="-0.015572,0.13435"/>
946          <Shape Type="disk" Radius="0.13153" MaterialId="human_naked" Position="0.008758,0.05895"/>
947          <Shape Type="disk" Radius="0.1375" MaterialId="human_naked" Position="0.013628,0"/>
948          <Shape Type="disk" Radius="0.13153" MaterialId="human_naked" Position="0.008758,-0.05895"/>
949          <Shape Type="disk" Radius="0.09565" MaterialId="human_naked" Position="-0.015572,-0.13435"/>
950       </Agent>
951       <Agent Type="pedestrian" Id="4" Mass="78.0" Height="1.725" MomentOfInertia="1.63" FloorDamping="4.50" AngularDamping
952       ↪ ="4.50">
953          <Shape Type="disk" Radius="0.09391" MaterialId="human_naked" Position="-0.01529,0.156092"/>
954          <Shape Type="disk" Radius="0.12914" MaterialId="human_naked" Position="0.0086,0.068492"/>
955          <Shape Type="disk" Radius="0.135" MaterialId="human_naked" Position="0.01338,2e-06"/>
956          <Shape Type="disk" Radius="0.12914" MaterialId="human_naked" Position="0.0086,-0.068498"/>
957          <Shape Type="disk" Radius="0.09391" MaterialId="human_naked" Position="-0.01529,-0.156088"/>
958       </Agent>
959    </Agents>
```

`AgentDynamics.xml` file

```
961
962  <Agents>
963      <Agent Id="0">
964          <Kinematics Position="0.000,0.000" Velocity="0.00,0.00" Theta="0.0" Omega="0.0"/>
965      <Dynamics Fp="0.00,0.0" Mp="0.0"/></Agent>
966      <Agent Id="1">
967          <Kinematics Position="0.279,0.030" Velocity="0.00,0.00" Theta="0.0" Omega="0.0"/>
968      <Dynamics Fp="0.00,0.0" Mp="0.0"/></Agent>
969      <Agent Id="2">
970          <Kinematics Position="0.692,-0.067" Velocity="0.00,0.00" Theta="0.0" Omega="0.0"/>
971      <Dynamics Fp="0.00,0.0" Mp="0.0"/></Agent>
972      <Agent Id="3">
973          <Kinematics Position="1.070,0.037" Velocity="0.00,0.00" Theta="0.0" Omega="0.0"/>
974      <Dynamics Fp="0.00,0.0" Mp="0.0"/></Agent>
975      <Agent Id="4">
976          <Kinematics Position="1.448,0.012" Velocity="0.00,0.00" Theta="0.0" Omega="0.0"/>
977      <Dynamics Fp="0.00,0.0" Mp="0.0"/></Agent>
978  </Agents>
```

`AgentInteractions.xml` file for $t = 2.65\,\text{s}$

```
980
981  <?xml version="1.0" encoding="utf-8"?>
982  <Interactions>
983      <Agent Id="0">
984          <Agent Id="1">
985              <Interaction ParentShape="2" ChildShape="2" TangentialRelativeDisplacement="6.12494e-08,-5.00732e-07" Fn="
986          ↪ -42.4307,-5.19011" Ft="-0.734183,6.00217" />
987          </Agent>
988      </Agent>
989  </Interactions>
```

# D   Packing algorithm within the streamlit app

The `pack_agents_with_forces` method, detailed in Algorithm 1, simulates the arrangement of agents within a bounded environment by iteratively applying physics-inspired, force-based interactions to resolve overlaps and enforce boundary constraints. Additionally, a temperature-based cooling mechanism is used to gradually reduce the magnitude of rotation, helping the system to stabilise. The algorithm relies on the following forces:

**Agent-agent repulsive force**

For every pair of agents $i$ and $j$, a repulsive force is computed that decays exponentially with

---

**Algorithm 1:** Agent packing with a force-based algorithm

---

1 **Method** `pack_agents_with_forces`(*repulsion_length, desired_direction, variable_orientation*):

2     **foreach** *agent* **do**

3         RotateTo(agent, *desired_direction*)

4     **end**

5     $T \leftarrow 1.0$                            ▷ Initial temperature

6     **for** *iteration* ← 1 **to** *MAX_NB_ITERATIONS* **do**

7         **foreach** *agent i* **do**

8             **forces** $\leftarrow [0,0,0]$                ▷ [x, y, rotation]

9             **foreach** *agent* $j \neq i$ **do**

10                 **forces**$_{\mathrm{xy}}$ += `repulsive_force`(*i, j, repulsion_length*)   ▷ $\mathbf{F}_{i,j}^{\mathrm{rep}}$

11                 **if** *overlap(i, j)* **then**

12                     **forces**$_{\mathrm{xy}}$ += `contact_force`(*i, j*)      ▷ $\mathbf{F}_{i,j}^{\mathrm{contact}}$

13                     **forces**$_{\mathrm{rot}}$ += `rotational_force`(*T*)      ▷ $f^{\mathrm{rot}}$

14                 **end**

15             **end**

16             **if** *boundary exists* **and** *(agent i is in contact with* **or** *outside the boundary)* **then**

17                 **forces** += `boundary_forces`(*i, T*)       ▷ $\mathbf{F}^{\mathrm{bound}}$

18             **end**

19             **if** *variable_orientation* **then**

20                 $\theta_i \leftarrow \theta_i +$ **forces**$_{\mathrm{rot}}$           ▷ Update orientation

21             **end**

22             $\mathbf{r}_{\mathrm{new}} = \mathbf{r}_{\mathrm{current}} +$ **forces**$_{\mathrm{xy}}$ **if** *valid_position(*$\mathbf{r}_{\mathrm{new}}$*)* **then**

23                 $\mathbf{r}_i \leftarrow \mathbf{r}_{\mathrm{new}}$                ▷ Update position

24             **end**

25         **end**

26         $T \leftarrow \max(0, T - 0.1)$               ▷ Cooling

27     **end**

---

the distance between their centroids:

$$\mathbf{F}_{i,j}^{\mathrm{rep}} = \begin{cases} e^{-|\mathbf{r}_i-\mathbf{r}_j|/\lambda}\, \dfrac{\mathbf{r}_i-\mathbf{r}_j}{|\mathbf{r}_i-\mathbf{r}_j|} & \text{if } |\mathbf{r}_i-\mathbf{r}_j| > 0 \\ \text{random small vector} & \text{otherwise} \end{cases} \tag{D.1}$$

where $\mathbf{r}_i$ is the centroid of agent $i$ and $\lambda$ is the repulsion length.

**Contact force**

If two agents' shapes overlap, a contact force is applied to push them apart:

$$\mathbf{F}_{i,j}^{\mathrm{contact}} = \begin{cases} k\, \dfrac{\mathbf{r}_i-\mathbf{r}_j}{|\mathbf{r}_i-\mathbf{r}_j|} & \text{if } |\mathbf{r}_i-\mathbf{r}_j| > 0 \\ \text{random small vector} & \text{otherwise} \end{cases} \tag{D.2}$$

where $k$ is the contact intensity.

**Rotational force**

When rotational dynamics are enabled, a random angular adjustment is applied, scaled by the

temperature $T$ of the system:

$$f^{\text{rot}} = \text{Uniform}(-\alpha, \alpha) \cdot T \tag{D.3}$$

where $\alpha$ is the maximum rotational intensity.

**Boundary forces**

If an agent is outside the boundaries or in contact with a force $\mathbf{F}^{\text{bound}}$ is computed to push it back inside, expressed as the sum of:

- ⋆ a contact force (as above) between the agent's centroid and the closest point on the boundary of the agent's centroid.
- ⋆ a rotational force (as above), scaled by the current temperature.

# E   Contributing

To ensure that our open-source platform, written in Python and C++, remains high quality and easy to maintain, we rely on a **continuous integration (CI)** pipeline that runs a series of automated checks on every contribution submitted via a GitHub pull request. The complete contribution workflow is described in detail in the `CONTRIBUTING.md` file in the repository, which is written to be accessible even to contributors who are not familiar with GitHub. All checks can be executed locally during development; the same checks are also run automatically on every pull request by the *pre-commit.ci* service and by *GitHub Actions* on both `macos-latest` and `ubuntu-latest` runners. To run all these tests locally from the project root, execute:

```
uv run pre-commit run --all-files
cd ./tests/mechanical_layer
./run_mechanical_tests.sh
```

These automated checks fall into two broad categories: (i) ***style and quality checks***, which enforce formatting, coding conventions, documentation rules, and basic static analysis; and (ii) ***functional checks***, which run tests to verify that the numerical behaviour of the functions is correct. For the Python code, we use:

- ⋆ **Ruff [58]:** This tool performs both *linting* and *formatting*. It therefore detects common mistakes and violations of established Python practices, such as logical errors, overly complex functions, undeclared variables, and deprecated constructs. It also automatically formats the code (indentation, spacing, and comments), keeping the Python interface consistently structured and easy to read.
- ⋆ **Mypy [59]:** This static type checker verifies that the types of variables and function signatures are used consistently throughout the code. It checks that the runtime usage of variables is compatible with the declared type hints in the code and in the function documentation, helping to catch errors where an incorrect value is passed, returned, or propagated.
- ⋆ **NumPydoc validation [60]:** This hook ensures that all public Python functions and classes have docstrings that follow a clear and standardised NumPy-style format.
- ⋆ **Pytest:** This tool runs a comprehensive suite of unit and integration tests on the Python wrapper (including both configuration files generation and mechanical layer tests) and on the Jupyter notebooks. Any unexpected behaviour or failing test is immediately reported.

For the C++ files, we use:

* **clang-format [61]:** This tool formats the C++ source code according to the formatting rules recommended by Google. In particular, it enforces consistent indentation, spacing, and line breaks, and it places curly brackets according to the *Allman* style.
* **clang-tidy [62]:** This static analysis tool examines the C++ code to catch common programming mistakes and potential bugs before execution. It identifies issues such as violations of coding style, incorrect use of interfaces (for example, calling functions with incompatible arguments), and type-related problems that can be detected purely from the source code.
* **cpplint [63]:** This tool checks that the C++ code adheres to the full set of coding style guidelines recommended by Google, complementing `clang-format` with higher-level style rules (for example, file organisation, naming conventions, and header usage).

For the shell scripts, we use **shfmt [64]** to format them uniformly. To detect and correct common misspelt words across the whole project, we use the **CodeSpell [65]** tool. Rather than checking against a full dictionary, it targets a curated list of frequent typographical errors. We also use the **nbqa [66]** tool in Jupyter notebooks, and our own scripts to check for Doxygen documentation errors and check that the copyright headers are present and correctly formatted across all project files.

# Acronyms

| | |
|---|---|
| ANSURII | ANthropometric SURvey 2. 3, 5, 6, 11, 13, 15, 17 |
| CoM | Center of Mass. 22, 24, 25 |
| EORCA | Elliptical Optimized Reciprocal Collision Avoidance. 4 |
| NHANES | National Health and Nutrition Examination Surveys. 6 |
| US | United States of America. 5, 6 |
| VHP | Visible Human Project. 5 |

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
