# Peer review of "LEMONS: An open-source platform to generate non-circuLar, anthropometry-based pEdestrian shapes and simulate their Mechanical interactiONS in two dimensions"

_SciPost Physics Codebases_

## Round 2 · Author Response

Dear Editors, dear Reviewers,
We have taken due consideration of the detailed reports of the two Referees, who should be thanked for their careful reading of our manuscript and their knowledgeable comments.
We were glad to read that overall the Reviewers find our work worthy of interest, even though they deem some changes absolutely necessary. We will see in detail how we have implemented these changes and revised our manuscript in the following, but before that we should clarify the scope of our work in relation with two main issues identified by the Reviewers:
-
Realism of the model and the (bottleneck) simulation: It is critical for us to stress that the model and code that we openly release is not a full pedestrian dynamics model. It is designed to produce realistic shapes in two dimensions and to solve the dynamics when physical contacts occur, but not to model the decisional process leading to the selection of a (desired) velocity, i.e., in what direction each agent intends to move at a given time. We will put forward in the future a pedestrian model that combines such a decisional layer with the mechanical model of LEMONS, but the reason why we chose to release LEMONS as a stand-alone open-source code is to enable modellers to easily include realistic physical shapes and contact handling in two dimensions in their own (decisional) model.
In the case study of the original manuscript, we had illustrated how LEMONS works with the example of a bottleneck flow, with the implementation of an (explicitly admitted) very crude decisional layer. Both Reviewers have observed that the results with this crude decisional layer lacks realism to some extent. We agree that this choice of scenario (where the decisional layer strongly affects the outcome) was not very wise to illustrate our model. Instead, following the Reviewers' advice, we now show the propagation of a mechanical push through a row of people, which almost exclusively hinges on the mechanical layer and, therefore, compares much more favourable with experimental data, as expected.
-
Force-based or velocity-based dynamics: The mathematical form of our dynamical equation has led to the (wrong) impression that the dynamics we describe are inertial, i.e., correspond to a 'force-based model'. As a matter of fact, this equation was just Newton's second law, which holds for the mechanical interactions in any case. Depending on the value of the relaxation time
$$t^{\text{(transl)}}$$that enters this equation, the dynamics will be either inertial (underdamped, large$$t^{\text{(transl)}}$$) or overdamped (small$$t^{\text{(transl)}}$$). The value we selected in the case study leads to rather overdamped dynamics, which do not display unrealistic oscillations of agents.
Below, we address one by one the comments of Reviewers 1 and 2.
Sincerely,
The Authors.

---

## Round 2 · List of Changes



---

## Editorial Decision

unknown